# Benchmarking at the Edge of Comprehension

**Samuele Marro** [1 2]  **Jialin Yu** [1 2]  **Emanuele La Malfa** [1 2]  **Oishi Deb** [1]  **Jiawei Li** [3]  **Yibo Yang** [4]  **Ebey Abraham** [5]
**Sunando Sengupta** [5]  **Eric Sommerlade** [5]  **Michael Wooldridge** [1 2]  **Philip Torr** [1 2]

## Abstract

As frontier Large Language Models (LLMs) increasingly saturate new benchmarks shortly after they are published, benchmarking itself is at a juncture: if frontier models keep improving, it will become increasingly hard for humans to generate discriminative tasks, provide accurate ground-truth answers, or evaluate complex solutions. If benchmarking becomes infeasible, our ability to measure any progress in AI is at stake. We refer to this scenario as the *post-comprehension regime*. In this work, we propose Critique-Resilient Benchmarking, an adversarial framework designed to compare models even when full human understanding is infeasible. Our technique relies on the notion of *critique-resilient correctness*: an answer is deemed correct if no adversary has convincingly proved otherwise. Unlike standard benchmarking, humans serve as bounded verifiers and focus on localized claims, which preserves evaluation integrity beyond full comprehension of the task. Using an itemized bipartite Bradley-Terry model, we jointly rank LLMs by their ability to solve challenging tasks and to generate difficult yet solvable questions. We showcase the effectiveness of our method in the mathematical domain across eight frontier LLMs, showing that the resulting scores are stable and correlate with external capability measures. Our framework reformulates benchmarking as an adversarial generation-evaluation game in which humans serve as final adjudicators.

[1]University of Oxford [2]Institute for Decentralized AI [3]Tsinghua University [4]King Abdullah University of Science and Technology [5]Microsoft. Correspondence to: Samuele Marro <samuele.marro@eng.ox.ac.uk>.

*Proceedings of the $43^{rd}$ International Conference on Machine Learning*, Seoul, South Korea. PMLR 306, 2026. Copyright 2026 by the author(s).

## 1. Introduction

Benchmarking makes the progress of AI capabilities measurable and underpins nearly every claim of improvement of Large Language Models (LLMs) (Hendrycks et al., 2020; Liang et al., 2022; Achiam et al., 2023; Chiang et al., 2024). However, as frontier models acquire stronger reasoning capabilities, benchmarks have struggled to keep up and are often criticized for being static (Kiela et al., 2021), ambiguous in what they measure (Bean et al., 2025), and ephemeral. The last aspect, i.e., the fact that benchmarks quickly saturate, is the most concerning and threatens to erode our trust in benchmarking itself. The trajectory of math reasoning benchmarks is exemplary in this sense. In just five years, benchmarks evolved from grade-school arithmetic (GSM8K Cobbe et al., 2021) to competition math (MATH Hendrycks et al., 2021). They then reached Olympiad problems (AIME) and finally advanced to research-level questions (FrontierMath Glazer et al., 2024) that demand days of expert work. Each generation requires more specialized expertise to create, more effort to verify, and saturates faster as models improve. Similar patterns appear for multimodal reasoning (Phan et al., 2025).

For the first time in the history of the field, we are witnessing a paradigm shift where humans can no longer reliably generate *verified*, challenging questions and ground-truth answers. This scenario, which we refer to as the **post-comprehension regime**, makes it extremely challenging to compare the capabilities of different LLMs, as doing so relies on our ability to generate and verify frontier problems. While delegating question-and-answer generation and evaluation to LLMs seems sufficient, it is only a necessary condition that must be complemented by a human-in-the-loop verification of correctness. For humans who verify a model's output, the generated "ground truth" provides no epistemic foundation for evaluation on its own. A core challenge in the post-comprehension regime thus involves establishing that AI-generated solutions are correct through methods that humans can ultimately trust.

This paper introduces **Critique-Resilient Benchmarking,**[1]

---

[1]Our code is available at `https://github.com/idai-institute/critique-resilient-benchmarking`.

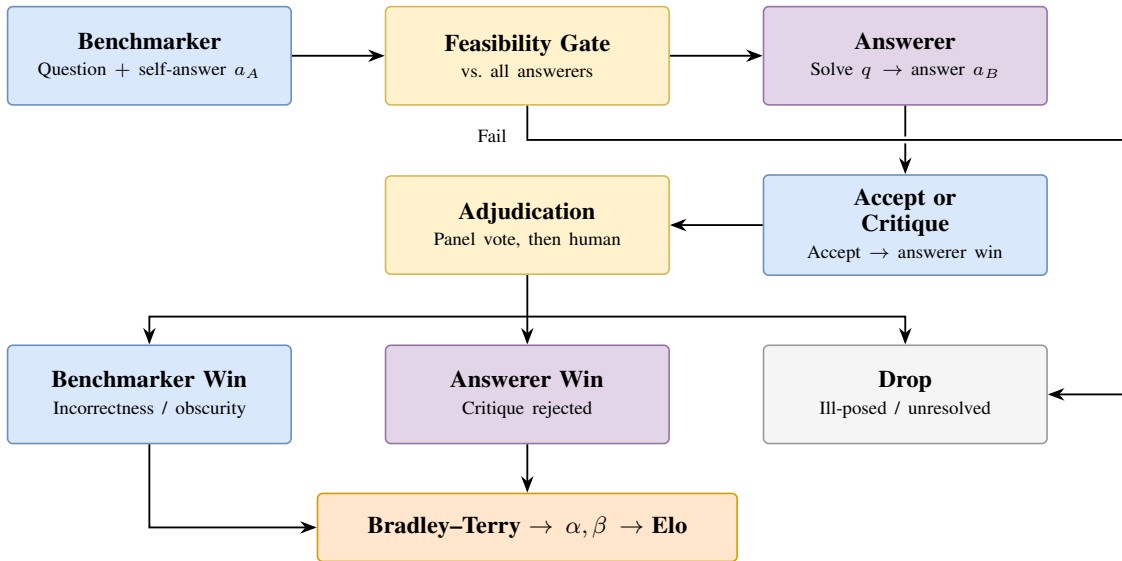

*Figure 1.* Overview of Critique-Resilient Benchmarking. Feasibility gating admits a question only if its self-answer $a_A$ survives critique from all other models and no ill-posedness critique is upheld (Section 4). The claim type determines the outcome: an upheld incorrectness or obscurity critique is a benchmarker win, an upheld ill-posedness critique drops the question, and a rejected (or absent) critique is an answerer win. Only win/loss episodes enter the itemized bipartite Bradley-Terry fit, which yields answerer strength $\beta$ and benchmarker strength $\alpha$.

a protocol designed for the post-comprehension regime that replaces ground-truth grading with **critique-resilient correctness**. Given a task and an answer generated by an LLM, an answer is considered correct only if it survives adversarial evaluation by other critic models attempting to produce local, verifiable claims of error. In this context, correctness is formalized as resistance to falsification, and the presence of the oracle (i.e., humans) scales because it is promoted to the role of validator of an automated LLM-to-LLM process.

The approach leverages an asymmetry present in domains such as computer science and math. While verifying the correctness of an entire solution may be infeasible, checking if a *specific alleged error* occurred remains tractable and sound. A counterexample to a claimed lemma, a failing test case, or an algebraic mistake are all examples of localized witnesses that can be checked independently. One may argue that this paradigm trades soundness for completeness (i.e., the absence of an error does not imply the correctness of an answer). In practice, LLM benchmarks already rely on imperfect proxies (Bean et al., 2025); our move makes the failure mode explicit and auditable. We propose relegating humans to a role that is compatible with increasing difficulty, without abandoning their epistemic role. Humans, as **bounded verifiers**, adjudicate narrow claims (e.g., "Does this counterexample actually refute the lemma?") rather than certifying global correctness.

Furthermore, our model addresses the problem of assigning a difficulty label to each question. Within our paradigm, we model task complexity as a property of the LLM's interac-

tion. We introduce an **itemized bipartite Bradley-Terry model** (Bradley & Terry, 1952) to estimate two latent parameters for each system: *answerer strength* (the ability to produce critique-resilient solutions) and *benchmarker strength* (the ability to pose hard-but-solvable questions and identify errors). This reframes benchmark design as a measurable capability, subject to explicit guardrails (feasibility gating and claim compliance rules) that prevent adversarial or ill-posed tasks.

We test our protocol in the mathematical domain and validate it across eight frontier and non-frontier LLMs. Our approach ranks LLMs by their ability to produce challenging question/answer pairs and debate about their correctness. Results show that model scores are stable under resampling and correlate well with traditional, human-designed benchmarks. Furthermore, rankings remain consistent even when weaker models replace humans as adjudicators, indicating that bounded verification is robust to wider gaps between the models and the adjudicator.

**Conflict of Interest Disclosure**   The authors EA, SS, and ES are employed by Microsoft, which leads the development of Phi-4, which was among the ones evaluated in this paper. SM received funding (directly or indirectly) from Microsoft, OpenAI (developers of GPT-5.2 and GPT-3.5), Google (developers of Gemini 3 Pro), and Anthropic (developers of Claude 4.5 Opus).

## 2. Benchmarking Beyond Human Capability

A benchmark is a suite of tests designed to quantify, in measurable and comprehensive terms, how a system responds to a *stimulus*. With LLMs and, more generally, in supervised machine learning, benchmarks assess the capabilities of a model with pairs of inputs/output examples drawn from a target distribution. Such pairs are typically (i) produced by humans (aided by algorithmic techniques, including LLMs Wang et al., 2023), (ii) correct by design (the output is the ground truth the model is expected to give), and (iii) consistent (i.e., each input/output pair is encoded in a standard format and tests a precise set of competencies that is determined by humans at design time). Historically, this design methodology has held well enough to make the paradigm (dataset → answer key → grading) a standard in the machine learning and LLM community.

However, the growing capabilities of frontier AI systems threaten the foundations of such a paradigm along every dimension. Consider the trajectory of math benchmarks: it took roughly three years for an LLM (Gemini 1.5 Pro Team et al., 2024) to surpass 90% accuracy on GSM8K (Cobbe et al., 2021). The MATH benchmark (Hendrycks et al., 2021), released in 2021, followed a similar trajectory, with GPT-o1, which in December 2024 achieved almost perfect accuracy on it (Jaech et al., 2024). While the complexity of new math datasets increases, and so does the time required to develop them, frontier models now exhibit very high performance at release time. AIME 2025 problems, released in February 2024, were solved on arrival by Grok-3, scoring 93.3% (xAI, 2025). Even math benchmarks where a single instance requires hours to days to be solved by human experts, such as FrontierMath (Glazer et al., 2024), have shown rapid improvement by models (OpenAI, 2025, released in November 2024, as of December 2025, GPT-5.2 Pro achieves a score of 40.7% on its Tiers 1–3 and 31.3% on Tier 4 questions).

Regardless of memorization concerns (Singh et al., 2024), it is evident that the ephemerality of these benchmarks calls into question the sustainability of such a paradigm.

### 2.1. The Post-Comprehension Regime

We define the *post-comprehension regime* as the situation where the current paradigms of benchmarking are not effective at measuring any significant improvement in frontier LLMs. In such a regime, one or more of the following conditions are true:

**(A1)** Humans cannot reliably generate frontier questions.
**(A2)** Ground truths are unavailable: even if a model proposes a reference answer, humans cannot practically verify it.
**(A3)** Holistic evaluation of answers is infeasible.

**(A4)** Difficulty labels stop being meaningful, as humans cannot distinguish genuine challenge from ambiguity or ill-posedness.

While the use of tools and formal verification can mitigate these challenges, they do not eliminate them: their use still relies on understanding their behavior and verifying that the inputs are well-specified. We further elaborate on this point in Appendix A.

### 2.2. Bounded Verification of Localized Claims

Despite (A2) and (A3), evaluation can still be feasible. The key observation is that many domains admit *short, checkable certificates of failure* even when global correctness is beyond reach.

Consider the difference between confirming that a 50-page proof is entirely valid versus checking whether a specific step follows from the previous, given an alleged error. The first task may be infeasible; the second is often tractable.

**Definition 2.1** (Witness-admitting domain). A task domain is *witness-admitting* if incorrect answers admit *witnesses of incorrectness*, i.e., objects $w$ that certify an error and can be verified using only a local excerpt of the answer.

Examples include counterexamples to claimed lemmas, algebraic errors in specific steps, failing test cases, violated invariants, and internal contradictions. In each case, verifying the witness requires examining only a local excerpt, not the complete solution. We stress that some domains lack this structure (subjective evaluation, existential claims without constructive proofs), and that our framework applies best where the witness-admitting property holds broadly.

The asymmetry between global verification and local witness checking has been studied theoretically in the context of debate protocols: for example, Brown-Cohen et al. (2024) shows that a broad class of problems admits doubly-efficient debate protocols in which a polynomial-time verifier can adjudicate disputes between provers despite being unable to solve the underlying problem directly. Their results provide formal grounding for the intuition behind Definition 2.1: bounded verification of localized claims is tractable across a much wider range of problems than those admitting end-to-end checking.

**Definition 2.2** (Verifier). A *verifier* $V$ is an agent with budget $B$ and procedure

$$\text{Check}_V(w; q, a, B) \in \\ \{\text{UPHELD}, \text{REJECTED}, \text{UNRESOLVED}\}, \quad (1)$$

which determines whether a witness $w$ is valid for a question-answer pair $(q, a)$ within budget $B$. We require that verifiers are sound (i.e., they do not uphold invalid witnesses).

This structure yields an asymmetric but honest guarantee: if

a witness is verified, the answer is genuinely incorrect; if no witness is produced, we know only that the answer survived scrutiny, not that it is correct. In the post-comprehension regime, this is the strongest guarantee available.

## 2.3. Design Requirements

The four assumptions described in Section 2.1 imply four requirements for viable evaluation:

**Requirement 1** (R1). **Delegated question generation.** Questions must be produced by models. The protocol must detect and filter ill-posed or adversarially unanswerable questions.

**Requirement 2** (R2). **No trusted answer keys.** Correctness must be defined pragmatically through observable outcomes.

**Requirement 3** (R3). **Localized verification.** Judgments must decompose into verifying narrow, specific claims. Verifiers adjudicate bounded witnesses, not global correctness.

**Requirement 4** (R4). **Endogenous difficulty.** Difficulty must be inferred from observed outcomes rather than annotated in advance.

Section 3 addresses R2 and R3 by defining *critique-resilient correctness*. Section 4 specifies the protocol, including feasibility gating for R1. Section 5 addresses R4 via a statistical model that treats difficulty as latent.

# 3. Critique-Resilient Correctness

Section 2.1 established that witness-admitting domains allow localized verification of incorrectness claims. This section translates that insight into a concrete notion of correctness suitable for the post-comprehension regime: an answer is accepted if no adversary can produce a verified witness of failure within fixed budgets.

## 3.1. From Witnesses to Critiques

In our protocol, witnesses are surfaced through *critiques*, i.e., structured claims produced by critic models targeting a question-answer pair.

**Definition 3.1** (Critique). Given $(q, a)$, a *critique* is a claim $c$ produced by a critic model, together with a witness $w$ sufficient for a verifier to evaluate the claim.

**Definition 3.2** (Valid critique). Fix a verifier $V$ and budget $B$. A critique $(c, w)$ targeting $(q, a)$ is *valid* if

$$\text{Check}_V(w; q, a, B) = \texttt{UPHELD}. \tag{2}$$

We distinguish three claim types, each corresponding to a different failure mode:

- **Incorrectness claim:** The answer contains a specific error. The witness $w$ might be a counterexample, a violated constraint, a failing test, or an internal contradiction.
- **Ill-posedness claim:** The question itself is invalid, i.e., underspecified, inconsistent, or ambiguous in a way that changes the answer. The witness demonstrates the issues, such as two incompatible interpretations, contradictory constraints, or a critical undefined term.
- **Obscurity claim:** The answer cannot be verified within budget because critical steps or assumptions are missing, unexplained, or stated at insufficient precision for correctness to be checked with reasonable effort. The witness identifies specific gaps that preclude verification (e.g., a major claim asserted without justification, a symbol used without prior definition, or a computational proof that cannot be checked in sufficient time).

## 3.2. Critique-Resilient Correctness

We now define correctness without ground truth, relative to (i) a class of critics and (ii) a bounded verification process.

**Definition 3.3** (Critique-resilient answer). Let $\mathcal{C}$ be a set of critic models and $(V, B)$ a verifier with a given budget. An answer $a$ to question $q$ is *critique-resilient* against $\mathcal{C}$ under $(V, B)$ if, after running the protocol's critique-and-resolution procedure with critics in $\mathcal{C}$, no valid incorrectness or obscurity critique is produced against $(q, a)$.

Equivalently: $a$ is accepted if it survives adversarial attempts to produce a verified witness of failure within budget.

This notion of correctness is explicitly relative to the critics, the verifier, and the budget. A stronger set of critics or a more capable verifier might surface errors that weaker ones miss. This relativity makes it explicit that "correctness" in the post-comprehension regime means being resistant to the best available falsification attempts, instead of relying on an omniscient oracle.

## 3.3. The Evaluation Game

Critique-resilient correctness reframes evaluation as an adversarial interaction between two roles:

- **Benchmarker (questioner + critic):** Authors questions and attempts to falsify answers by producing valid critiques.
- **Answerer (solver + defender):** Attempts to produce critique-resilient answers.

This framing makes *benchmark design itself a measurable capability*. A strong benchmarker produces questions that cause other models to fail while remaining well-posed, and can localize errors in faulty answers. A weak benchmarker produces questions that are easily solved or ill-posed, and misses errors in incorrect answers.

**Definition 3.4** (Evaluation game outcome). Fix a benchmarker $A$ and answerer $B$. An interaction produces a question $q$ (authored by $A$) and an answer $a_B$ (by $B$). Under the protocol:

- $B$ **wins** if $a_B$ is critique-resilient: no valid incorrectness or obscurity critique is upheld.
- $A$ **wins** if $B$ fails to answer, or if a valid incorrectness or obscurity critique against $a_B$ is upheld.
- The outcome is **dropped** if an ill-posedness critique against $q$ is upheld, or if the protocol cannot resolve the relevant claims within its adjudication budget.

An upheld ill-posedness critique invalidates $q$ rather than rewarding either party: since the question itself is defective, its latent difficulty $\theta_{a,i}$ has no meaningful interpretation, and including such traces in the Bradley-Terry fit would contaminate the latent space.

## 4. The Protocol

This section specifies the evaluation procedure that instantiates critique-resilient correctness. Each episode involves a **benchmarker** $A$ (that generates questions and critiques answers) and an **answerer** $B$ (that attempts to provide solutions). Figure 1 illustrates the complete flow.

### 4.1. Procedure Overview

The protocol proceeds in two stages:

**Stage 1: Feasibility Gating.** The benchmarker produces a question $q$ along with a proposed answer $a_A$. To prevent adversarial or ill-posed questions, $a_A$ must itself be critique-resilient: the answerer may critique it, and if the critique is upheld in the adjudication process, the question is marked invalid and dropped. Similarly, the answerer can produce an ill-posedness claim with witness, which, if upheld, leads to dropping. In practice, a question $q$ is admitted if and only if (i) no ill-posedness critique against $q$ is upheld, and (ii) the benchmarker's self-answer $a_A$ survives critique attempts from every other model in the pool. A single upheld critique of any kind is sufficient to invalidate $q$.

**Stage 2: Adversarial Evaluation.** If the question passes feasibility gating, the answerer produces either an answer $a_B$ or a failure declaration. If $B$ fails, $A$ wins. Otherwise, $A$ must either accept $a_B$ as critique-resilient or produce a critique $(c, w)$ targeting it. Acceptance yields an immediate answerer win; critiques proceed to adjudication.

### 4.2. Adjudication

The framework is agnostic to the specific adjudication process. We instantiate one version: after an optional debate phase where claimant and defender clarify the disputed claim, a panel of judge models (excluding those involved in the episode) vote `UPHELD` or `REJECTED`. Unanimous automated verdicts stand; otherwise, the claim escalates to human adjudicators who review the specific claim, its witness, and the debate transcript. If humans return `UNRESOLVED`, the episode is dropped.

### 4.3. Outcomes

Each episode resolves to one of three outcomes:

- **Answerer wins:** $B$ answers and $A$ accepts, or $A$'s incorrectness or obscurity critique is rejected.
- **Benchmarker wins:** $B$ fails to answer, or $A$'s incorrectness or obscurity critique is upheld.
- **Drop:** an ill-posedness critique against $q$ is upheld, adjudication returns `UNRESOLVED`, or required artifacts are missing.

If an interaction proves that a question is invalid (due to being ill-posed or having an incorrect self-answer), all traces involving that question are dropped.

## 5. From Outcomes to Rankings

Section 4 yields a set of evaluation traces, each resolving whether an answer is critique-resilient under bounded verification. This section converts the resolved traces into two interpretable rankings (namely, answering strength and benchmarking strength) without requiring difficulty labels.

### 5.1. Eligible Episodes and Outcome Encoding

Index question authors by $a \in \{1, \ldots, A\}$, answerers by $b \in \{1, \ldots, B\}$, and let $q_{a,i}$ denote the $i$-th question written by author $a$. For each attempt where $b$ answers $q_{a,i}$, the protocol returns one of:

- **Answerer wins:** $b$ produces an answer accepted as critique-resilient.
- **Benchmarker wins:** $b$ fails to answer or a critique against $b$'s answer is upheld.
- **Drop:** unresolved adjudication, missing artifacts, or question invalidation (e.g., upheld ill-posedness or failure of the feasibility gate).

We fit rankings only on *eligible binary outcomes*. Let $\mathcal{D}$ be the set of eligible edges $((a, i), b)$. For each interaction, define $y_{(a,i),b} \in \{0, 1\}$, with $y_{(a,i),b} = 1$ for answerer win and $y_{(a,i),b} = 0$ for benchmarker win. Dropped outcomes are excluded from $\mathcal{D}$ and analyzed separately.

## 5.2. Itemized Bipartite Bradley-Terry Model

We model each eligible interaction as a Bradley-Terry game between an answerer $b$ and a question instance $q_{a,i}$:

$$\Pr\big(y_{(a,i),b} = 1\big) = \sigma(\beta_b - \theta_{a,i}), \qquad \sigma(t) = \frac{1}{1 + e^{-t}}, \tag{3}$$

where $\beta_b$ is *answerer strength* and $\theta_{a,i}$ is the latent *difficulty* of question $q_{a,i}$.

Since questions are authored by models, we decompose difficulty into an author-level component plus an item residual:

$$\theta_{a,i} = \alpha_a + \delta_{a,i}. \tag{4}$$

Substituting yields the model used throughout:

$$\Pr\big(y_{(a,i),b} = 1\big) = \sigma(\beta_b - \alpha_a - \delta_{a,i}). \tag{5}$$

Here $\alpha_a$ is *benchmarker strength* (the ability to produce questions that induce failures, conditional on passing the feasibility gate and spotting the failures), and $\delta_{a,i}$ captures within-author variation.

**Why bipartite.** Besides the one we used, another possible setup involves three roles: authoring questions, answering them, and generating critiques. We considered this tripartite decomposition and abandoned it for identifiability reasons: a scenario in which $B$'s answer survives critique is observationally equivalent under two distinct explanations (the question was easy, or the critic missed a real error), and these explanations imply opposite updates to the question-author and critic parameters. For a more in-depth explanation, refer to Appendix B.

**Identifiability.** Only differences $\beta_b - \alpha_a - \delta_{a,i}$ affect the likelihood, so parameters are defined up to a global shift. We fix this with a centering constraint $\sum_{b=1}^{B} \beta_b = 0$.

**Estimation and regularization.** The interaction graph can be sparse and near-separable, so we fit by MAP with Gaussian priors:

$$\beta_b \sim \mathcal{N}(0, \sigma_\beta^2), \quad \alpha_a \sim \mathcal{N}(0, \sigma_\alpha^2), \quad \delta_{a,i} \sim \mathcal{N}(0, \sigma_\delta^2). \tag{6}$$

This setting is equivalent to an L2-regularized logistic regression on $\mathcal{D}$. The prior scales $(\sigma_\beta, \sigma_\alpha, \sigma_\delta)$ control shrinkage; we select them by empirical Bayes (Appendix G).

**Reporting uncertainty.** We report uncertainty via a cluster bootstrap over questions (Appendix H), which preserves item-level dependence.

## 5.3. Interpretation

Scores can be interpreted as follows:

- High $\beta_b$: model $b$ frequently produces answers that survive critique and adjudication → strong *answering under bounded verification*.
- High $\alpha_a$: model $a$ frequently authors questions that cause others to fail while remaining feasible → strong *benchmark design* in the operational sense induced by the protocol.

For presentation, we map parameters affinely to an Elo-like scale (Appendix H.2); this preserves rankings and changes only units.

We stress that $\alpha_a$, while useful for interpreting experimental results, should not be treated as an intrinsic capability outside the evaluation setting: a model that produces harder questions is not necessarily a better model in any general sense. In contrast, $\beta_b$ has a clearer interpretation: a model that answers complex questions well is, other things equal, a more capable model.

# 6. Experiments

This section describes the experimental instantiation of the protocol, including domain, models, data generation, and evaluation procedure. Prompt templates and statistics about the generated dataset are deferred to Appendix D.

## 6.1. Experimental Setup

**Domain.** We focus on mathematics as a prototypical witness-admitting domain: solutions are often long and technical, end-to-end evaluation is expensive, and localized error claims (counterexamples, missing assumptions, invalid steps) are frequently checkable even when full verification is not. To cover a broad range of mathematical content, we use the MSC2020 top-level classification categories as topic prompts, excluding categories primarily focused on physics, computer science, and history. This leaves 44 core mathematics categories, each used as part of the prompt during question generation (refer to Appendix D for the full list).

**Models.** We evaluate eight LLMs spanning a range of capabilities and providers: GPT-3.5 (OpenAI, 2022), GPT-4o (OpenAI, 2024), Phi 4 (Abdin et al., 2024), Llama 4 Maverick (Meta AI, 2025), GPT-5.2 (OpenAI, 2025), Gemini 3 Pro Preview (Google, 2025), Claude 4.5 Opus (Anthropic, 2025), and DeepSeek v3.2 Speciale (Liu et al., 2025). Unless otherwise stated, we use default settings for provider APIs (a model-by-model configuration table is provided in Appendix D). The overall API cost of our experiments (including preliminary studies) was ∼3.3k USD.

**Dataset construction.** For each model $a$ and each MSC category $t$, we generate exactly one question package $\langle q_{a,t}, a_{a,t} \rangle$, consisting of a question and the author's pro-

posed answer. This yields 352 questions total (44 categories $\times 8$ models). For each authored question $q_{a,t}$, every other model $b \neq a$ attempts an answer, producing 2464 candidate question-answer pairs.

**Protocol instantiation.** We run CRB with the following choices:

- *Debate budget:* up to 5 turns per side, with early concession allowed.
- *Judges:* all models except the two involved in the current episode.
- *Adjudication outcomes*: To collect finegrained data, we divide adjudication outcomes into sub-outcomes. For example, we distinguish between REJECTED due to a claim being incorrect versus due to the claim being only about stylistic mistakes. Refer to Appendix D for a full list of sub-outcomes.
- *Resolution rule:* unanimous automated outcomes are accepted; otherwise the claim is human-adjudicated. For the purposes of our experiments, we count different outcome sub-categories as distinct.
- *Claim types:* incorrectness, ill-posedness, and obscurity; upheld obscurity counts as a benchmarker win.
- *Critique compliance:* if a critique contains at least one upheld material error, the benchmarker wins; otherwise the answerer wins. To prevent "shotgun" critiques with many low-quality subclaims, we treat excessive claim counts as grounds for rejection, though no such cases were observed in practice.

**Human adjudicators.** Since the protocol benchmarks frontier mathematical capabilities, high-quality human labels were the primary constraint. All human adjudicators hold Master's or PhD qualifications in Computer Science or Mathematics. They were allowed to use any tool (including LLMs) to assist with understanding the problem and were permitted to skip questions outside their competence. Low-confidence evaluations were reviewed by a second labeler; disagreements were resolved by a third labeler as tiebreaker. The labeling process required approximately 90 person-hours.

### 6.2. Research Questions

We organize the analysis around two research questions:

**RQ1 (Meaningful scores).** Does the protocol produce answerer scores $\hat{\beta}$ that behave like measures of underlying capability, i.e., do they generalize to new interactions and align with external capability signals?

*Rationale:* A reliable benchmarking protocol should yield scores that are internally consistent (stable under resampling) and externally valid (correlated with independent

| Benchmark | Spearman $\rho$ | Kendall $\tau$ |
|---|---|---|
| AIME 2025 | 0.851 [0.738, 0.952] | 0.706 [0.500, 0.857] |
| BRUMO 2025 | 0.830 [0.690, 0.952] | 0.673 [0.429, 0.857] |
| HMMT Feb 2025 | 0.819 [0.707, 0.934] | 0.662 [0.473, 0.837] |

*Table 1.* Spearman $\rho$ and Kendall $\tau$ (with 95% CIs) correlation measures between answerer Elo (bootstrapped) and mean external benchmarks.

| Metric | Baserate | Model | $\Delta$ vs baseline |
|---|---|---|---|
| Accuracy ↑ | 0.646 | 0.922 | +0.276 |
| Log-loss ↓ | 0.650 | 0.208 | -0.442 |
| Brier ↓ | 0.229 | 0.061 | -0.168 |

*Table 2.* Pooled test metrics compared to a baserate predictor.

measures of model capability). We assess internal consistency via bootstrap stability analysis and external validity by correlating $\hat{\beta}$ with published scores on established mathematics benchmarks.

**RQ2 (Weak models are consistent with humans).** Does the protocol allow weak models to reliably adjudicate outcomes, even when stronger models are being evaluated? Specifically, do the rankings change when weak models replace humans as final adjudicators?

*Rationale:* As frontier models improve, human adjudicators may struggle with increasingly complex claims. We cannot study superhuman evaluation directly, but we have a useful proxy: weak models. If a weak model (e.g., GPT-3.5), despite low answering performance, produces adjudication decisions that agree with human verdicts, it suggests that bounded verification can remain reliable even when the verifier is less capable than the systems being evaluated.

## 7. Results

**RQ1 (Meaningful scores).** We perform topic-based bootstrapping and compute means and 95% CIs for the beta scores, which we report (converted to Elo) in Figure 2 (questioner Elos can be found in Appendix E). As expected, answerer Elo correlates with model performance, with frontier models (DeepSeek v3.2 Speciale, Claude 4.5 Opus, GPT-5.2, Gemini 3 Pro Preview) forming a cluster with the highest performance, followed by smaller and older models (Llama 4 Maverick, Phi 4, GPT-4o) and finally by the weakest model, GPT-3.5. We empirically validate this intuition by computing correlations between answerer Elo and modern math benchmarks, namely AIME 2025, BRUMO 2025, and HMMT Feb 2025. The results, reported in Table 1, show that Elos are indeed monotonically correlated with traditional, human-generated benchmarks. Refer to Appendix E for scatter plots. We test internal consistency

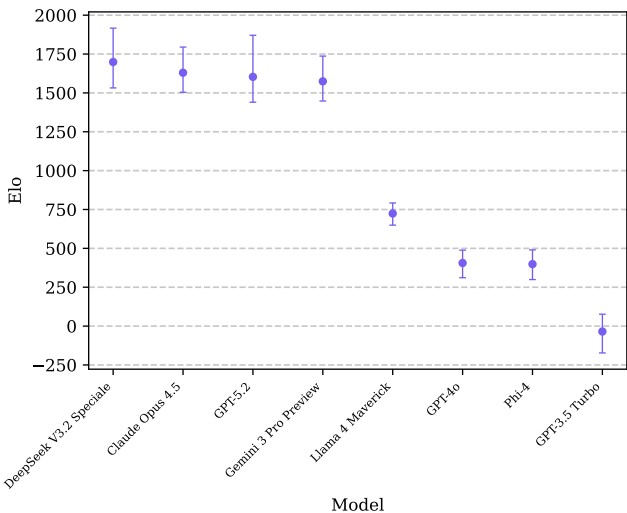

*Figure 2.* Answerer Elos with 95% CI intervals.

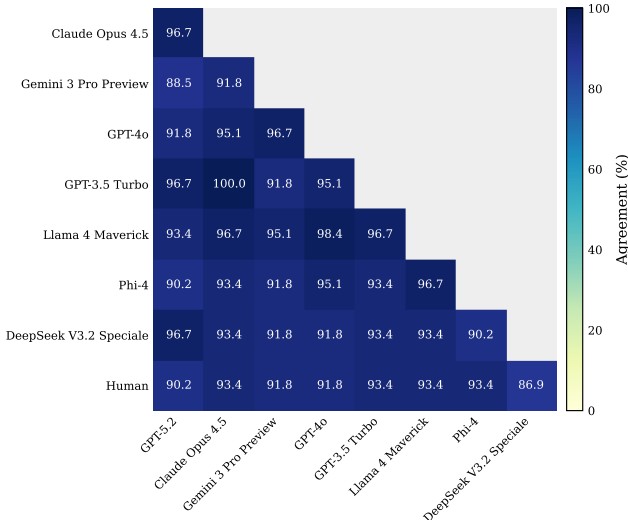

*Figure 3.* Agreement rates between human evaluators and models

by measuring predictive validity, i.e., whether the scores computed on a subset of the questions predict the outcome of the unseen questions. We perform 5-fold validation and report the results in Table 2. As the table shows, the test log-loss and Brier score are significantly lower compared to the baserate predictor. In other words, the scores on the train set strongly generalize to unseen examples, showing that the protocol is consistent. We report calibration curves in Appendix E.

**RQ2 (Weak models are consistent with humans).** We use each model as adjudicator (censoring identities of every model involved in the trace) and compare outcomes with human adjudications. As shown in Figure 3, all models (including weak ones such as GPT-3.5 and GPT-4o) consistently agree with the human adjudicator. We also instantiate the Bradley-Terry model using LLM-generated adjudications and find that scores and rankings are consistent (see Appendix E). This shows that, even when models such as GPT-3.5 and GPT-4o (which have significantly inferior answering and benchmarking performance compared to the models they are benchmarking) are tasked with adjudicating outcomes, they can do so very reliably ($\rho \in [0.98, 1], \tau \in [0.93, 1]$), which implies that, within CRB, adjudication is simpler than benchmark and answer generation. Per-sub-outcome agreement rates are reported in Figure 9.

**Additional analyses.** To complement our results, we conducted four additional analyses, reported in full in Appendix F. *(i) Critic reliability:* using GPT-5.4-high as a ground-truth proxy, we find that critique agreement rates are uniformly high across all models (0.88–0.96), while answer correctness rates vary widely (0.16–0.99). This shows that catching errors is substantially easier than avoiding them,

which makes CRB tractable in practice. *(ii) Mutual incompatibility:* among pairs of non-proven-wrong answers to the same question, only 4.23% (160/3779) are mutually incompatible per the GPT-5.4-high classifier; this is concentrated in pairs involving weak models (14–18% for GPT-3.5 pairs, 0–0.4% for frontier model pairs), suggesting the protocol's false-negative rate shrinks as models improve. *(iii) Prompt sensitivity:* a three-way prompt ablation shows that answerer rankings are highly stable across prompt design choices (Kendall $W = 0.952$, mean pairwise Spearman $\rho = 0.929$). *(iv) $\alpha$–$\beta$ correlation:* benchmarker and answerer strengths are moderately but not fully correlated (Spearman $\rho = 0.69$, Kendall $\tau = 0.47$), suggesting they reflect distinct capabilities of models.

## 8. Related Work

**LLM-as-a-Judge and Critique-Based Evaluation.** A growing line of work replaces human evaluation with LLM judges, motivated by cost and scalability. MT-Bench and Chatbot Arena popularized this for open-ended evaluation, documenting systematic biases such as verbosity and position preference (Zheng et al., 2023). G-Eval uses strong LMs as reference-free evaluators with structured rubrics (Liu et al., 2023), while subsequent work targets specific biases such as length (Dubois et al., 2024). Our setting differs in two ways: we target post-comprehension regimes where evaluators cannot assess end-to-end correctness, and we shift from holistic scoring to *claim-scoped adjudication*, where judges decide only whether a specific falsifiable claim is upheld, reducing susceptibility to stylistic biases.

**Debate and Scalable Oversight.** AI Safety via Debate proposed training agents through zero-sum games judged

by humans (Irving et al., 2018), with subsequent theoretical work establishing doubly-efficient debate protocols that allow bounded verifiers to adjudicate disputes between more capable provers (Brown-Cohen et al., 2024). Recent work studies whether weaker judges can reliably select correct outcomes when stronger models argue (Kenton et al., 2024), and multi-agent debate has been used to improve LLM evaluators (Chan et al., 2023). Our protocol shares this spirit: evaluation is an interaction trace rather than a key-based lookup. The main difference is the purpose: scalable oversight typically asks whether debate enables the supervision of stronger systems during training or deployment, while CRB uses the same structural property for evaluation, with the BT model providing a way to aggregate adversarial outcomes into capability rankings.

**Dynamic and Adversarial Benchmarks.** Multiple efforts address benchmark saturation through competitive or renewable evaluation. ZeroSumEval uses competition-based protocols with Elo-style aggregation (Khan et al., 2025); Qi Town evaluates LLMs via head-to-head game play (Zhou et al., 2025b); YourBench generates question sets on demand (Shashidhar et al., 2025). We target a distinct failure mode: the unavailability of trusted ground truth. Unlike game-based competitions, our tasks remain open-ended; unlike generation systems, we treat benchmarker quality as part of what is measured.

**Modeling Ability and Difficulty.** Using latent-variable models to disentangle item difficulty and model ability has a long psychometric tradition (e.g., Item Response Theory Embretson & Reise, 2000) recently adopted for LLM evaluation (Truong et al., 2025; Zhou et al., 2025a). Our model adapts this to a setting where items are authored by the models being evaluated, inducing a bipartite structure that estimates both answerer strength and question-author strength.

**Formal Verification.** Benchmarks like miniF2F evaluate theorem proving with formal verification in Lean and Metamath (Zheng et al., 2021), and platforms like LeanDojo support large-scale reproducible evaluation (Yang et al., 2023). We view formal verification as powerful but domain-limited: it provides strong guarantees where applicable but does not cover tasks with informal or partial specifications. Our protocol extends meaningful evaluation to such domains while leveraging certificate-like witnesses when available.

## 9. Conclusion

As frontier systems approach the limits of human end-to-end understanding, the traditional benchmarking paradigm breaks down. This paper formalized that breakdown as the post-comprehension regime and introduced Critique-

Resilient Benchmarking, a protocol designed to operate within it.

The core move is to replace ground-truth correctness with critique-resilient correctness: an answer is accepted not because it matches a trusted reference, but because no adversary can produce a verified witness of failure within budget. This reframes evaluation as an adversarial game rather than a lookup against an answer key, grounded in a structural property of domains like mathematics and code where local witnesses of error remain checkable even when global verification is intractable.

In CRB, humans remain in the loop as bounded verifiers, adjudicating specific claims rather than certifying entire solutions. Two further innovations address delegated question generation: feasibility gating filters adversarially unsolvable tasks, and an itemized Bradley-Terry model infers difficulty from outcomes, yielding separate rankings for answering and benchmarking strength.

Our experiments suggest the approach is viable: rankings are stable, correlate with external benchmarks, and remain consistent when weaker models replace humans as adjudicators.

**Limitations and Outlook.** The framework assumes witness-admitting domains, which is a hypothesis that only some domains fully satisfy (though many useful ones do, including algorithmic problem solving and mechanism design). Additionally, critique-resilient correctness is explicitly relative: an answer that survives today's critics may fall to tomorrow's. We view this as a feature rather than a bug: unlike traditional benchmarks, which require new human-authored problems to remain discriminative, CRB can be re-run against stronger critics on the existing question pool, with the Bradley-Terry fit absorbing the updated outcomes. Finally, future regimes where bounded verification itself becomes infeasible would require fundamentally different approaches. Nevertheless, the deeper lesson is that evaluation does not require full comprehension: it suffices to recognize, when shown, that something specific is wrong. As models grow more capable, organizing evaluation around verifiable failure evidence may be not only useful but necessary.

## Impact Statement

This work addresses the challenge of evaluating AI systems as their capabilities approach or exceed human-level performance in specialized domains. We discuss both potential benefits and risks.

**Potential Benefits.** Reliable evaluation is foundational to AI safety and governance. As models become more capa-

ble, the inability to measure their performance undermines our ability to detect capability gains, compare systems, and make informed deployment decisions. By providing a framework that remains meaningful even when end-to-end human verification is infeasible, Critique-Resilient Benchmarking may help maintain human oversight of advanced AI systems during a critical period of capability growth. The framework's emphasis on falsifiable claims and bounded verification also promotes transparency: instead of treating evaluation as a black-box comparison against (potentially complex) answer keys, it reveals the specific grounds on which judgments are made.

**Potential Risks.** Our framework relies on the assumption that witness-admitting domains allow meaningful local verification even when global understanding fails. If this assumption breaks down (e.g., if adversaries learn to construct errors that evade detection by bounded verifiers), the framework could provide false assurance about system capabilities. Additionally, by formalizing benchmark design as a measurable capability, we risk incentivizing models optimized for adversarial question generation in ways that could transfer to other contexts. We have attempted to mitigate this through feasibility gating (requiring benchmarkers to answer their own questions), but acknowledge that adversarial dynamics in evaluation warrant continued scrutiny.

**Limitations of Scope.** Our experimental validation focuses on mathematics, a domain with relatively clear correctness criteria. The framework's applicability to domains with more subjective or contested notions of correctness (e.g., ethical reasoning, creative writing, policy analysis) remains an open question. We caution against overextending the approach to settings where the witness-admitting property does not hold.

## Acknowledgments

We would like to thank Leslie Tao and Andy Lee for their help. Samuele Marro is supported by the EPSRC Centre for Doctoral Training in Autonomous Intelligent Machines and Systems n. EP/Y035070/1, in addition to Microsoft Ltd. The Institute for Decentralized AI is supported by a grant by the AI Safety Fund.

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

# A. Why Benchmarking Breaks in the Post-Comprehension Regime

We now elaborate on why each assumption is likely to hold as AI capabilities advance.

## A.1. Why A1 is likely.

The supply of discriminative questions depends on the existence of tasks that (i) current frontier models cannot reliably solve, (ii) have well-defined correct answers, and (iii) can be authored by humans with reasonable effort. As models improve, condition (i) increasingly requires problems at or beyond the frontier of human expertise. However, problems at this frontier are expensive to produce: FrontierMath reportedly required months of effort from research mathematicians to generate a few hundred problems. Furthermore, the most capable human experts are a finite resource. Even crowdsourcing at scale (which has been applied successfully for datasets such as ImageNet (Russakovsky et al., 2015) or SQuAD (Rajpurkar et al., 2016)) fails when the required expertise is rare and the verification problem (A2) makes quality control intractable.

The alternative of procedural generation (as in some coding and mathematical benchmarks) works only for domains with formal specifications and scales poorly to open-ended reasoning. Attempts to increase difficulty by increasing problem length or complexity often introduce ambiguity faster than they increase genuine challenge, violating condition (ii).

## A.2. Why A2–A3 are likely.

For ground truth and holistic evaluation, two traditional approaches are often proposed: tool-assisted verification and formal verification.

**Tool-assisted verification.** Tool-assisted verification encompasses any aid (calculators, symbolic algebra systems, search engines, or AI assistants) that augments human evaluators. While such tools extend the boundary of what humans can verify, they do not eliminate it. Tool-assisted verification introduces new failure modes: the human must correctly operate the tool, interpret its outputs, and trust its implementation. As solutions grow in length and complexity, the cognitive burden of orchestrating verification eventually exceeds human capacity regardless of tool support.

**Formal verification.** Formal verification, i.e., using proof assistants like Lean, Coq, or Isabelle to machine-check correctness, provides strong guarantees where applicable. However, formal verification faces three fundamental limitations as a general solution to post-comprehension evaluation:

- **Coverage:** Many important tasks resist formalization. Even within mathematics, converting an informal proof to a formal one requires substantial expertise and effort, often exceeding the effort of producing the informal proof in the first place.
- **Specification correctness:** Formal verification checks that a solution satisfies a specification, but the specification itself may be wrong.
- **Scalability:** Current proof assistants require extensive human guidance to verify nontrivial results. While AI systems for formal mathematics are improving rapidly, using AI to generate formal proofs does not solve the evaluation problem: it merely shifts the question to "how do we verify the AI's formal proofs?" If the proofs are short enough for human inspection, they could have been verified informally; if they are too long, we are forced to trust the AI system we were trying to evaluate.

We therefore view formal verification as a valuable tool that complements but does not replace the need for evaluation methods that work when full verification is infeasible.

## A.3. Why A4 is likely.

Difficulty labeling requires humans to make comparative judgments: Is problem A harder than problem B? By how much? Such judgments are reliable when humans have experience solving similar problems and can estimate the cognitive effort required. At the frontier, this calibration breaks down. A problem may appear difficult because it requires a novel insight, because it is underspecified in ways the labeler does not notice, because it assumes background knowledge the labeler lacks, or because it is simply impossible. Distinguishing these cases requires solving (or nearly solving) the problem, which is precisely what we cannot assume humans can do.

## B. Unipartite vs. Bipartite vs. Tripartite Decomposition

Section 5 fits a bipartite Bradley-Terry model with two latent parameters per system: answerer strength $\beta_b$ and benchmarker strength $\alpha_a$. Two natural alternatives are to decompose the benchmarker role further, separating *question authoring* from *error identification*, or to merge all capability scores into a single one. This appendix explains why we adopted the bipartite decomposition and provides evidence that the two retained roles capture distinct capabilities.

### B.1. The Identifiability Obstacle

Consider an episode in which $B$'s answer survives $A$'s critique attempts. Under a tripartite decomposition with separate parameters for authoring ($\alpha_a^Q$), answering ($\beta_b$), and critiquing ($\alpha_a^C$), this outcome admits two explanations:

- The question was easy: The correct update reduces $\alpha_a^Q$, increases $\beta_b$, and leaves $\alpha_a^C$ unchanged.
- The critic missed a real error: The correct update leaves $\alpha_a^Q$ unchanged, leaves $\beta_b$ unchanged (or reduces it, depending on convention), and reduces $\alpha_a^C$.

These two updates push the parameters in opposite directions, but the observable outcome is the same in both cases. Symmetrically, a benchmarker win admits an analogous pair of explanations (the question was hard, or the critic spotted an error that a weaker critic would not have identified), with the same observational equivalence.

Without an exogenous signal that distinguishes an easy question from a missed error, the likelihood cannot identify $\alpha_a^Q$ and $\alpha_a^C$ separately. The bipartite model resolves this by collapsing both into a single benchmarker parameter $\alpha_a$, which captures the joint capability of authoring questions that induce failures *and* surfacing errors when they occur.

### B.2. Partial Disentanglement

The identifiability obstacle is fundamental at the level of pairwise outcomes, but partial information can be recovered from *disagreements across critics*. For example, if GPT-5.2 upholds an incorrectness claim on an answer that GPT-3.5 accepted, this is informative about the gap in critic capability between the two systems, independent of the question's difficulty. Our framework exploits this implicitly through the panel of judges in the adjudication stage (whose verdicts inform the final outcome), but it does not feed it back into the BT fit as a separate parameter. A finer-grained model could, in principle, exploit such disagreements to partially identify a critic dimension; we leave this to future work.

### B.3. Why Not Unipartite?

Finally, one could consider collapsing all capabilities into a single score, as is more common in more traditional LLM evaluations. However, if question authoring and error identification were near-redundant capabilities, we would expect benchmarker strength $\alpha_a$ to be highly correlated with answerer strength $\beta_b$, since a model good at producing hard questions on a topic would presumably also be good at solving them. Empirically, the two are moderately but not fully correlated: we observed a Spearman $\rho = 0.69$ and Kendall $\tau = 0.47$ (computed from the point estimates in Figures 2 and 6). For example, Llama 4 Maverick ranks notably higher as an answerer than as a benchmarker, while GPT-4o shows the opposite pattern. This moderate correlation supports the bipartite decomposition: the two roles are distinct enough to warrant separate parameters, but not so independent that further decomposition would be statistically tractable.

## C. Sample Questions

To showcase the complexity of generated questions, we report 8 sample questions (one from each model).

### C.1. GPT-3.5 Turbo: Radical of an Ideal in $\mathbb{Z}[x]$

**Topic:** `commutative_algebra`

**Question.** Let $R$ be a commutative ring with unity and let $I$ be an ideal in $R$. Define the radical of $I$, denoted $\sqrt{I}$, as the set of all elements $r \in R$ such that there exists a positive integer $n$ satisfying $r^n \in I$.

Consider the ring $R = \mathbb{Z}[x]$ of polynomials with integer coefficients and the ideal $I = \langle x^2 - 4 \rangle$ generated by $x^2 - 4$. Determine the radical $\sqrt{I}$ of the ideal $I$ in $R$.

## C.2. GPT-4o: Quotients by Discrete Normal Subgroups

**Topic:** `topological_groups_and_lie_groups`

**Question.** Let $G$ be a Lie group and let $H$ be a discrete normal subgroup of $G$. Prove that the quotient group $G/H$ is a Lie group. Furthermore, determine the dimension of $G/H$ in terms of the dimension of $G$.

## C.3. GPT-5.2: Mixing and Entropy of a Toral Automorphism

**Topic:** `dynamical_systems_and_ergodic_theory`

**Question.** Let $\mathbb{T}^2 := \mathbb{R}^2/\mathbb{Z}^2$ be the 2-torus with Haar (Lebesgue) probability measure $m$. Let

$$A = \begin{pmatrix} 2 & 1 \\ 1 & 1 \end{pmatrix} \in SL(2, \mathbb{Z}), \qquad T : \mathbb{T}^2 \to \mathbb{T}^2, \quad T([x]) = [Ax],$$

where $[x]$ denotes the class of $x \in \mathbb{R}^2$ modulo $\mathbb{Z}^2$.

1) Prove that $m$ is $T$-invariant.

2) Prove that $T$ is mixing with respect to $m$, i.e. for all $f, g \in L^2(\mathbb{T}^2, m)$,

$$\int_{\mathbb{T}^2} f \circ T^n \, g \, dm \longrightarrow \int f \, dm \int g \, dm \quad \text{as } n \to \infty.$$

3) Compute the Kolmogorov–Sinai entropy $h_m(T)$ exactly.

## C.4. Claude 4.5 Opus: Singularities and Genus of a Projective Plane Curve

**Topic:** `algebraic_geometry`

**Question.** Let $C$ be the projective closure in $\mathbb{P}^2_{\mathbb{C}}$ of the affine plane curve defined by $y^3 = x^5 - x^4$.

(a) Find all singular points of $C$.

(b) Compute the geometric genus of $C$.

## C.5. Gemini 3 Pro Preview: Cohomology of the Real Grassmannian $G(2, 4)$

**Topic:** `manifolds_and_cell_complexes`

**Question.** Let $M = G(2, 4)$ be the real Grassmannian manifold consisting of all 2-dimensional linear subspaces of $\mathbb{R}^4$.

1. Calculate the Euler characteristic $\chi(M)$.

2. Compute the integral cohomology groups $H^k(M; \mathbb{Z})$ for all $k \geq 0$.

3. Determine the ring structure of the cohomology with coefficients in $\mathbb{Z}_2$, denoted $H^*(M; \mathbb{Z}_2)$, by providing a presentation in terms of generators and relations.

## C.6. Llama 4 Maverick: Adjoint Representations and Central Discrete Subgroups

**Topic:** `topological_groups_and_lie_groups`

**Question.** Let $G$ be a connected Lie group with Lie algebra $\mathfrak{g}$. Suppose that $\mathfrak{g}$ is simple and that $G$ has a discrete subgroup $\Gamma$ such that $G/\Gamma$ is compact. Prove that $\Gamma$ is contained in the center of $G$ if and only if the adjoint representation of $G$ on $\mathfrak{g}$ descends to a representation of $G/\Gamma$.

### C.7. Phi-4: Homology and Uniformization of a Closed Surface

**Topic:** `algebraic_topology`

**Question.** Let $X$ be a compact, connected, orientable surface of genus $g \geq 2$. Consider the universal cover $\tilde{X}$ of $X$, which is homeomorphic to the hyperbolic plane $\mathbb{H}^2$. Let $\pi_1(X)$ denote the fundamental group of $X$, and let $\rho : \pi_1(X) \to \mathrm{Isom}^+(\mathbb{H}^2)$ be the action of $\pi_1(X)$ on $\tilde{X}$ by deck transformations.

1. Determine the structure of the first homology group $H_1(X; \mathbb{Z})$.

2. Prove that the Euler characteristic $\chi(X)$ is related to the genus $g$ by the formula $\chi(X) = 2 - 2g$.

3. Using the action $\rho$, show that the quotient space $\mathbb{H}^2/\pi_1(X)$ is homeomorphic to $X$.

### C.8. DeepSeek V3.2 Speciale: Blow-Up Resolution of the Quadratic Cremona Map

**Topic:** `algebraic_geometry`

**Question.** Consider the rational map $\varphi : \mathbb{P}^2 \to \mathbb{P}^2$ over $\mathbb{C}$ given in homogeneous coordinates by

$$\varphi(x : y : z) = (yz : xz : xy).$$

Let $X$ be the blow-up of $\mathbb{P}^2$ at the three points

$$P_1 = (1 : 0 : 0), \quad P_2 = (0 : 1 : 0), \quad P_3 = (0 : 0 : 1).$$

Denote by $\pi : X \to \mathbb{P}^2$ the blow-down morphism, and let $E_1, E_2, E_3$ be the exceptional divisors over $P_1, P_2, P_3$ respectively. Let $H = \pi^* \mathcal{O}_{\mathbb{P}^2}(1)$ be the pullback of the hyperplane class.

(a) Show that $\varphi$ lifts to a morphism $\Phi : X \to \mathbb{P}^2$, i.e. $\Phi = \varphi \circ \pi$ is a morphism.

(b) Compute the class of the line bundle $\Phi^* \mathcal{O}_{\mathbb{P}^2}(1)$ in $\mathrm{Pic}(X)$ in terms of $H, E_1, E_2, E_3$.

(c) Prove that $\varphi$ is birational and determine its inverse rational map.

## D. Full Experimental Setup

**MSC2020 Areas** We report selected MSC2020 topics in Table 3.

**Model Configurations** We report model configurations in Table 4.

**Self-improvement loop.** To reduce trivial mistakes, we use an iterative self-evaluation loop for question generation, answer generation, and critique generation. For up to $K = 5$ iterations, the model (i) produces an artifact, (ii) self-critiques with a pass/fail decision, and (iii) if failed, revises by applying the critique. If no attempt passes after $K$ iterations, the artifact is marked as a failure.

**Dropped questions.** To improve coverage for weaker models, we allow up to 5 attempts to submit a valid (i.e., not failed, not ill-posed, and with a correct self-answer) question. Frontier models (GPT-5.2, Gemini 3 Pro, DeepSeek v3.2 Speciale, Claude 4.5 Opus) always produced a valid answer under this setting, while weaker models such as GPT-3.5 only managed to do so for 8 topics out of 44. We report question success rates in Table 6.

**Labelling UI** We report screenshots of our labelling UI in Figure 4.

**Summary statistics.** Out of 2464 pairs, after 5 attempts to generate valid questions, 1897 (76.99%) passed the feasibility gating stage. In 1819 cases (73.82%), Bob provided an answer, which was declared correct by Alice in 1201 (48.74%) and incorrect in 618 (25.08%) cases. The automated labellers upheld the claim in 566 (22.97%) cases, rejected the claim in 10 (0.41%) cases, and escalated to humans in 44 (1.79%) cases. Overall, out of 1897 traces used in the Bradley-Terry model, 671 (35.37%) were Alice victories, while 1226 (64.23%) were Bob victories. Besides incorrectness claims, humans also adjudicated 16 self-answer-incorrect claims and 1 illposedness claim (total: 61), and 125 claims for preliminary experiments and previous attempts to generate valid questions (overall total: 186). Refer to Figure 5 for a full breakdown of victory rates between different models.

**Hyperparameters** We report the hyperparameters (computed using the process described in Appendix G in Table 5). These values are used throughout all experiments, except for the predictive validity experiments (where they are estimated on each training split).

**Adjudication Outcomes** For claims about answers, we allow the following outcomes for automated judges and for adjudicators:

- `"claimant_wins"`: Alice's claim is correct: the answer has a substantive flaw that she correctly identified (any valid problem counts, even if she doesn't identify all flaws).

    - Use this even if Bob concedes, fixes the issue, or provides missing justification during the debate.
    - Bob's concession or fix **confirms** Alice was right.
    - Example: Alice correctly identifies that a theorem was applied without verifying hypotheses.
    - Treated as UPHELD.

- `"defender_wins_incorrect"`: Alice's claim is incorrect: Bob successfully demonstrates the answer is correct and Alice misidentified a problem that doesn't exist.

    - Use this when Alice claims an error exists but the mathematics is actually correct.
    - Example: Alice claims continuity wasn't proven, but Bob shows it was proven in line 3.
    - Treated as REJECTED.

- `"defender_wins_minor"`: Alice's claim is technically correct but about very minor issues only: Bob successfully shows the flaws Alice identified are very minor and don't affect mathematical correctness

    - Use this when Alice identifies only notational inconsistencies, minor presentation issues, etc.
    - Example: Alice points out inconsistent notation that doesn't affect understanding.
    - Treated as REJECTED.

- `"wrong_problem"`: There are issues with the answer, but Alice's specific diagnosis is incorrect or unfounded.

    - Use this when the answer has errors, but not the ones Alice identified.
    - Example: Alice says "the proof is circular" but actually a case is missing.
    - Treated as REJECTED.

- `"mixed"`: Alice makes multiple claims, some correct and some incorrect (not all claims are valid).

    - Only use for factually incorrect claims, not "all are correct, but some are nitpickings".
    - Example: Alice makes three critiques. One identifies a real error, but another makes a math mistake.
    - Treated as UPHELD.

- `"unknown"`: Cannot determine whether Alice's claim is valid with reasonable confidence.

    - Use when the debate doesn't provide enough clarity.
    - Use when both parties make valid but orthogonal points.
    - Treated as UNRESOLVED.

- `"other"`: A case not covered by the above options; specify in comments. Only available to final adjudicators.

– Treated as `UNRESOLVED`.

For ill-posedness claims, we use the following outcomes:

- `"claimant_wins"`: Alice's claim is correct: the question is ill-posed per the rubric (any valid problem counts, even if she doesn't identify all problems).

    – Use this even if Bob concedes or acknowledges the issue during the debate.
    – Example: Alice correctly identifies that the domain is unspecified.
    – Treated as `UPHELD`.

- `"defender_wins_incorrect"`: Alice's claim is incorrect: Bob successfully demonstrates the question is well-posed and Alice misidentified the issue.

    – Use this when Alice claims something is missing/ill-posed but it's actually present/correct.
    – Example: Alice claims domain is unspecified, but it's explicitly stated in the question.
    – Treated as `REJECTED`.

- `"wrong_problem"`: The question has issues, but Alice's specific diagnosis is incorrect or not substantiated.

    – Use this when the question is ill-posed, but not for the reason Alice claims.
    – Example: Alice says "the integral diverges" but actually the domain is undefined.
    – Treated as `REJECTED`.

- `"mixed"`: Alice makes multiple claims, some correct and some incorrect (not all claims are valid).

    – Only use for factually incorrect claims, not "all are correct, but some are nitpickings".
    – Example: Alice makes three claims about ill-posedness, but only one is actually valid.
    – Treated as `UPHELD`.

- `"unknown"`: Cannot determine whether Alice's claim is valid with reasonable confidence.

    – Use when the debate and available info don't provide enough clarity.
    – Use when both parties make valid but orthogonal points.
    – Treated as `UNRESOLVED`.

**Prompts**    We report our prompts in Figures 10 to 18. For the sake of brevity, generation guidances are reported in the code repository.

**External Benchmarks**    Since not all scores were available for each model, we used Matharena (Balunović et al., 2025) to recompute external benchmark scores with CIs. We report our results in Table 7.

## E. Additional Results

**Victory Rates**    We report victory rates in Figure 5.

**Questioner Elos**    We report questioner Elos in Figure 6. As stated in Section 5.3, we stress that questioner Elos should not be interpreted outside the context of the protocol.

**Scatter Plots**    We report in Figure 7 the scatter plots showing approximately monotonic correlation between answerer Elo and external benchmarks.

**Calibration Curves**    We report the calibration curves for our predictive validity experiment in Figure 8.

**Elo Consistency**    We report Spearman $\rho$ and Kendall $\tau$ between Elos computed using LLM adjudications and human adjudications in Tables 8 and 9.

### Question

Let $\mathcal{H}$ be a separable Hilbert space, and let $T : \mathcal{H} \to \mathcal{H}$ be a compact, self-adjoint operator. Consider the spectral decomposition of $T$ given by $T = \sum_{n=1}^{\infty} \lambda_n \langle \cdot, e_n \rangle e_n$, where $\{e_n\}$ is an orthonormal basis of $\mathcal{H}$ consisting of eigenvectors of $T$, and $\{\lambda_n\}$ are the corresponding eigenvalues.

1. Prove that the sequence $\{\lambda_n\}$ converges to zero.
2. Determine the conditions under which the series $\sum_{n=1}^{\infty} |\lambda_n|^2$ converges.
3. Explain the significance of this convergence in the context of the operator $T$.

> Copy Question

### Answer

1. **Convergence of $\{\lambda_n\}$ to zero:**

   Since $T$ is a compact operator on a Hilbert space $\mathcal{H}$, by the spectral theorem for compact operators, the non-zero eigenvalues $\{\lambda_n\}$ of $T$ can only accumulate at zero. This means that for any $\epsilon > 0$, there are only finitely many eigenvalues $\lambda_n$ such that $|\lambda_n| \geq \epsilon$. Therefore, $\lambda_n \to 0$ as $n \to \infty$.

2. **Convergence of $\sum_{n=1}^{\infty} |\lambda_n|^2$:**

   The series $\sum_{n=1}^{\infty} |\lambda_n|^2$ converges if and only if $T$ is a Hilbert-Schmidt operator. A compact operator $T$ is Hilbert-Schmidt if and only if the sum of the squares of its singular values (which are the absolute values of its eigenvalues in the case of a self-adjoint operator) is finite. Thus, $\sum_{n=1}^{\infty} |\lambda_n|^2 < \infty$ if and only if $T$ is Hilbert-Schmidt.

3. **Significance of the convergence:**

   The convergence of $\sum_{n=1}^{\infty} |\lambda_n|^2$ indicates that $T$ is a Hilbert-Schmidt operator. This is significant because Hilbert-Schmidt operators form an important class of compact operators with additional structure. Specifically, they are more "regular" than general compact operators, and they have a well-defined trace norm given by $\|T\|_{\mathrm{HS}} = \left(\sum_{n=1}^{\infty} |\lambda_n|^2\right)^{1/2}$. This norm provides a measure of the "size" of the operator in a way that is analogous to the $L^2$ norm for functions. Additionally, Hilbert-Schmidt operators are integral operators with square-integrable kernels, which connects them to various applications in functional analysis and quantum mechanics.

> Copy Answer

### Critique

Parts (1) and (2) are essentially correct: for compact self-adjoint $T$, nonzero eigenvalues can only accumulate at $0$, hence (with any enumeration of eigenvalues, including multiplicity) $\lambda_n \to 0$; and $\sum_n |\lambda_n|^2 < \infty$ is equivalent to $T$ being Hilbert-Schmidt (since for self-adjoint operators the singular values are $|\lambda_n|$).

However, part (3) contains substantive inaccuracies. It states: "they have a well-defined trace norm given by $\|T\|_{\mathrm{HS}} = \left(\sum |\lambda_n|^2\right)^{1/2}$," but this is not the trace norm; it is the Hilbert-Schmidt norm. The trace (nuclear) norm corresponds to $\sum_n s_n(T)$ (and for suitable self-adjoint/positive cases, $\sum_n |\lambda_n|$), not the square-sum. It also claims categorically that "Hilbert-Schmidt operators are integral operators with square-integrable kernels," which is not true on an arbitrary separable Hilbert space (it holds in specific $L^2$-type realizations, not in general). These errors make the explanation of the significance incorrect as written.

> Copy Critique

### Debate

- **Bob**: Thank you for your critique, Alice. Let's address the points you've raised:

1. **Convergence of $\{\lambda_n\}$ to zero:** Your observation is correct. I stated that the non-zero eigenvalues of a compact operator can only accumulate at zero, which implies $\lambda_n \to 0$ as $n \to \infty$. This part is indeed correct.

2. **Convergence of $\sum_{n=1}^{\infty} |\lambda_n|^2$:** You are right that this series converges if and only if $T$ is a Hilbert-Schmidt operator. I correctly identified the condition for $T$ being Hilbert-Schmidt.

3. **Significance of the convergence:** I acknowledge the inaccuracies in my explanation. The norm $\|T\|_{\mathrm{HS}} = \left(\sum_{n=1}^{\infty} |\lambda_n|^2\right)^{1/2}$ is indeed the Hilbert-Schmidt norm, not the trace norm. The trace norm is given by $\sum_n |\lambda_n|$ for suitable self-adjoint operators. Additionally, while Hilbert-Schmidt operators can be represented as integral operators with square-integrable kernels in specific $L^2$ settings, this is not universally true for all separable Hilbert spaces. I appreciate your pointing out these errors.

I concede that the explanation of the significance in part (3) contained inaccuracies and should be corrected to reflect the proper definitions and context.

*(a)* Problem trace

### Automated evaluations

- Judge `google-gemini-3-pro-preview` → **claimant_wins** (conf 5): Alice correctly identified that the answer misidentified the Hilbert-Schmidt norm ($\ell^2$ sum of eigenvalues) as the trace norm ($\ell^1$ sum of singular values). This is a substantive definitional error distinguishing two different operator ideals. Bob conceded this error in the debate.

- Judge `meta-llama-llama-4-maverick` → **claimant_wins** (conf 5): Alice correctly identified substantive inaccuracies in part (3) of Bob's answer. Bob conceded these errors, acknowledging that the norm discussed is the Hilbert-Schmidt norm, not the trace norm, and that the characterization of Hilbert-Schmidt operators as integral operators with square-integrable kernels is not universally true. These concessions confirm Alice's critique is valid.

- Judge `anthropic-claude-opus-4.5` → **claimant_wins** (conf 5): Alice correctly identified two substantive errors in part (3): (1) the answer incorrectly labels the Hilbert-Schmidt norm as the 'trace norm' (the trace/nuclear norm is ∑|λ_n|, not (∑|λ_n|²)^(1/2)), and (2) the claim that Hilbert-Schmidt operators are integral operators with square-integrable kernels is only true in L² realizations, not on arbitrary separable Hilbert spaces. Bob explicitly conceded both points in round 1, acknowledging these as inaccuracies. These are substantive mathematical errors—confusing two distinct norms and making an overly general claim—not minor stylistic issues.

- Judge `openai-gpt-4o-2024-08-06` → **claimant_wins** (conf 5): Alice correctly identified substantive inaccuracies in part (3) of Bob's answer. Bob conceded that the explanation of the significance of the convergence contained errors, specifically misidentifying the Hilbert-Schmidt norm as the trace norm and incorrectly stating that Hilbert-Schmidt operators are integral operators with square-integrable kernels in all separable Hilbert spaces. These are substantive flaws, confirming Alice's critique is valid.

- Judge `deepseek-deepseek-v3.2-speciale` → **claimant_wins** (conf 5): Alice correctly identified substantive errors in part (3) of the answer: the norm defined is the Hilbert-Schmidt norm, not the trace norm, and the claim that Hilbert-Schmidt operators are integral operators with square-integrable kernels is not generally true on an arbitrary separable Hilbert space. Bob conceded these errors. These are not minor issues; they affect the correctness of the answer. Therefore, Alice's critique is valid.

- Judge `openai-gpt-3.5-turbo-0125` → **defender_wins_incorrect** (conf 4): Bob correctly acknowledges the inaccuracies in his explanation of the significance of the convergence in part (3). He admits the errors regarding the Hilbert-Schmidt norm and the representation of Hilbert-Schmidt operators as integral operators with square-integrable kernels. By addressing and conceding these inaccuracies, Bob demonstrates that Alice's critique about the explanation in part (3) being incorrect is valid.

> Copy automated evaluations

> Copy all fields

*(b)* Automated evaluations

### Verdict

○ claimant_wins - Alice's claim is correct; the answer has a substantive flaw she identified.
○ defender_wins_incorrect - Alice's claim is incorrect; the answer is actually correct.
○ defender_wins_minor - Alice identified only minor issues that do not affect correctness.
○ wrong_problem - The answer has issues, but not the one Alice claimed.
○ mixed - Alice made multiple claims; some are correct and some are incorrect.
● unknown - Not enough evidence to judge Alice's claim.
○ other - Use for cases not covered; explain in comments.

Confidence (1-5)

1 — — — — — 3 — — — — — 5

Comments

> Save

> Guidance - Question

> Guidance - Answer

> Guidance - Critique

> Guidance - Evaluation

*(c)* Labelling options

*Figure 4.* Screenshots of the labelling interface for adjudication.

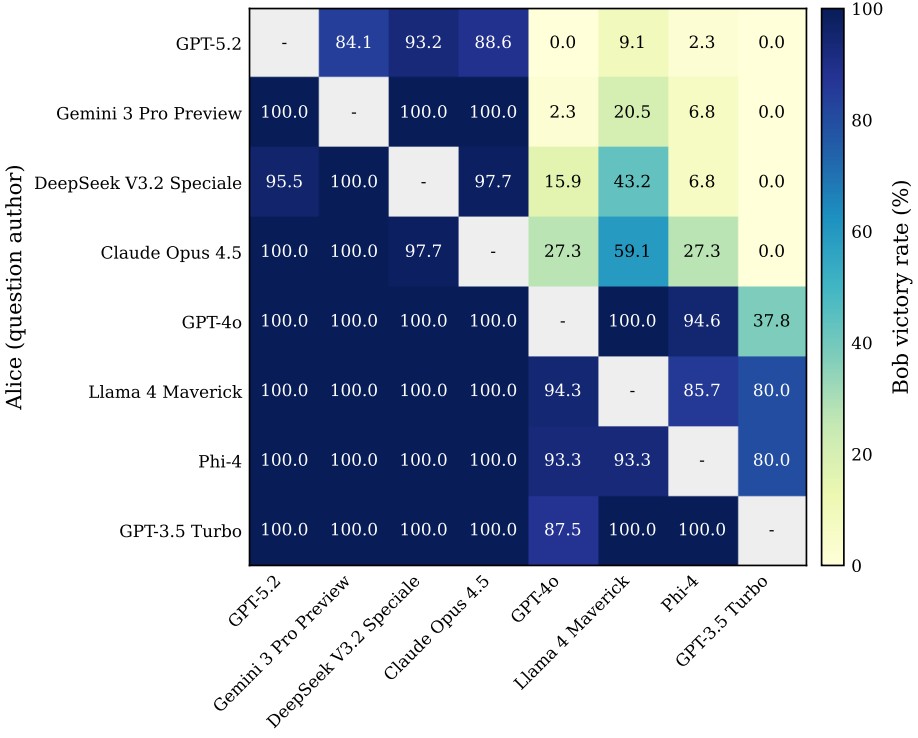

*Figure 5.* Victory rates among valid traces.

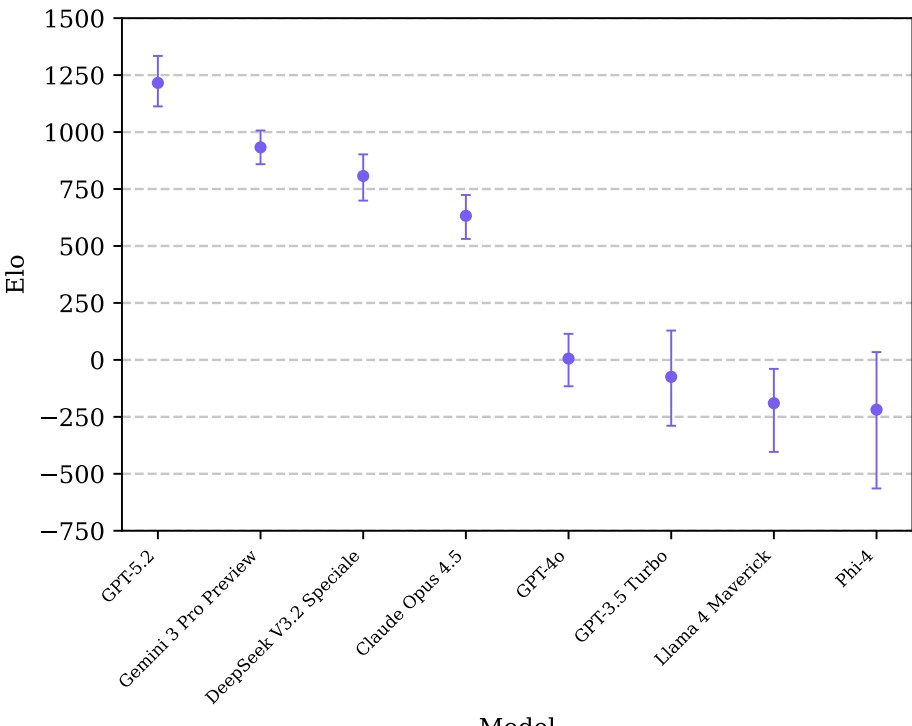

*Figure 6.* Questioner Elos with 95% CI intervals.

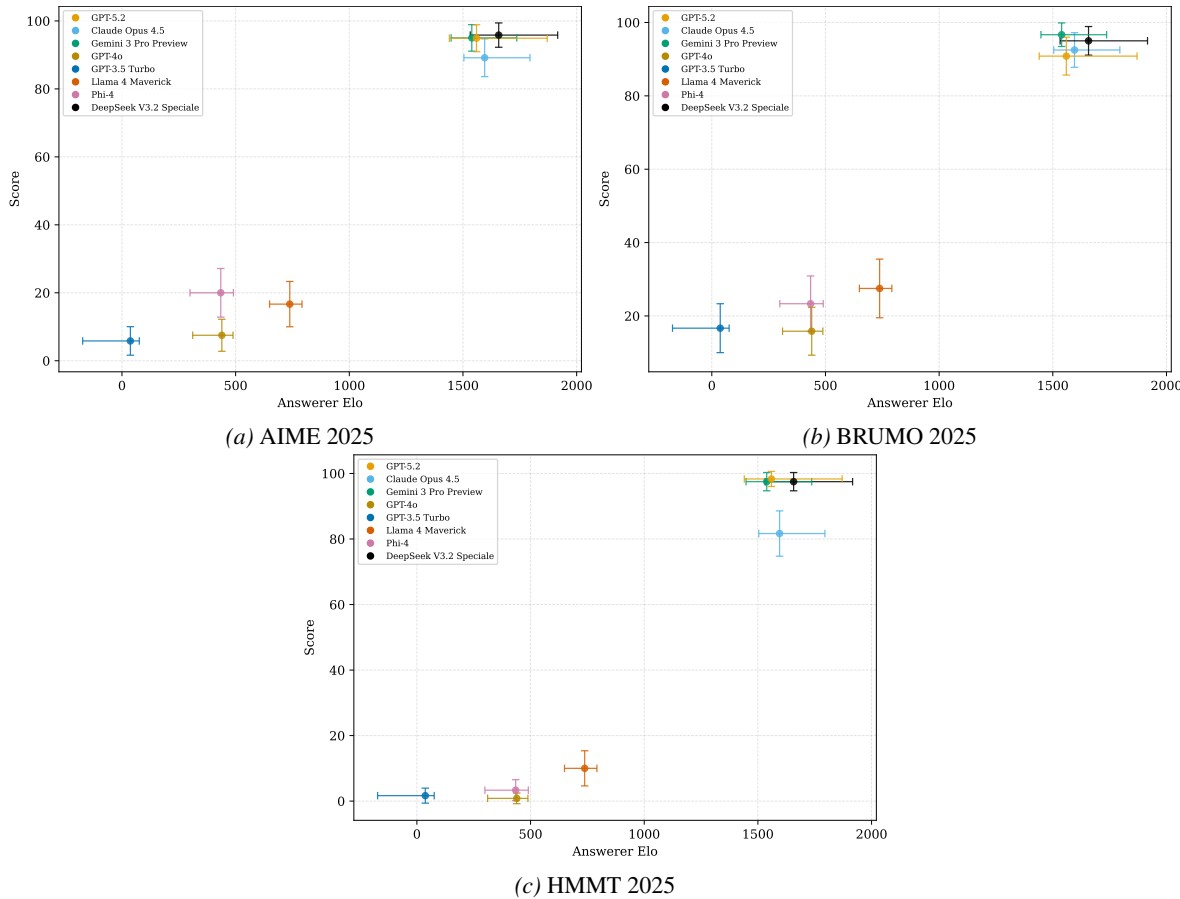

*(a)* AIME 2025

*(b)* BRUMO 2025

*(c)* HMMT 2025

*Figure 7.* Scatter plots for answerer Elo and external benchmarks, with 95% CIs for both Elo and external scores.

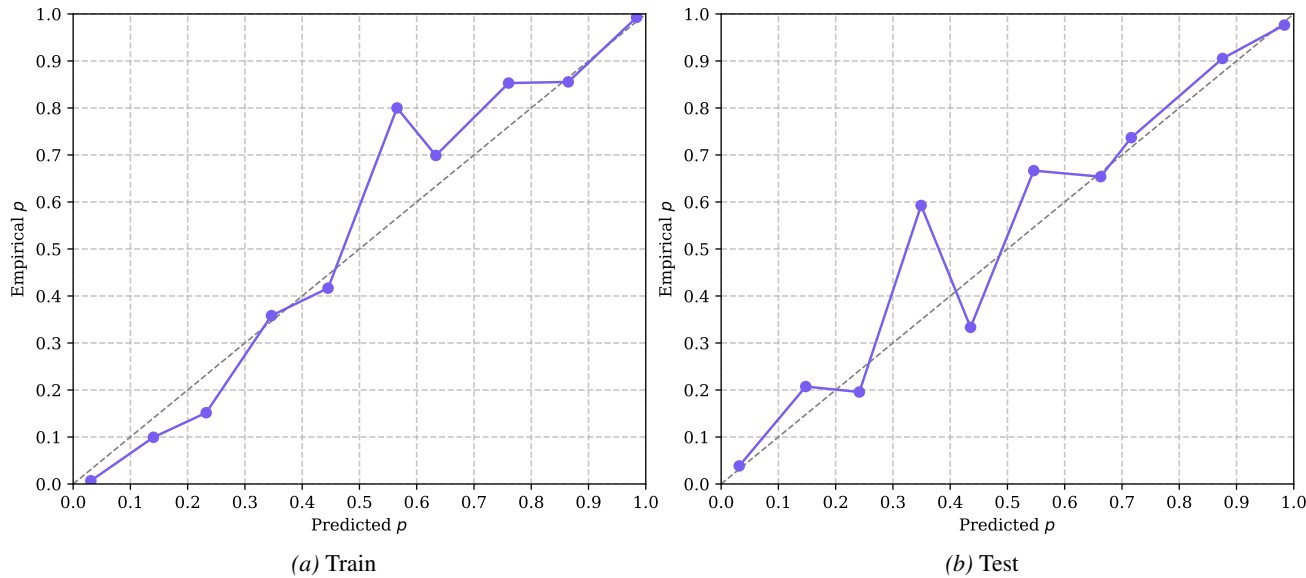

*(a)* Train

*(b)* Test

*Figure 8.* Calibration curves for our predictive validity experiment.

| Code | Topic | In? | Code | Topic | In? |
|---|---|---|---|---|---|
| 00 | General and overarching topics; collections | ✗ | 44 | Integral transforms, operational calculus | ✓ |
| 01 | History and biography | ✗ | 45 | Integral equations | ✓ |
| 03 | Mathematical logic and foundations | ✓ | 46 | Functional analysis | ✓ |
| 05 | Combinatorics | ✓ | 47 | Operator theory | ✓ |
| 06 | Order, lattices, ordered algebraic structures | ✓ | 49 | Calculus of variations and optimal control... | ✓ |
| 08 | General algebraic systems | ✓ | 51 | Geometry | ✓ |
| 11 | Number theory | ✓ | 52 | Convex and discrete geometry | ✓ |
| 12 | Field theory and polynomials | ✓ | 53 | Differential geometry | ✓ |
| 13 | Commutative algebra | ✓ | 54 | General topology | ✓ |
| 14 | Algebraic geometry | ✓ | 55 | Algebraic topology | ✓ |
| 15 | Linear and multilinear algebra; matrix theory | ✓ | 57 | Manifolds and cell complexes | ✓ |
| 16 | Associative rings and algebras | ✓ | 58 | Global analysis, analysis on manifolds | ✓ |
| 17 | Nonassociative rings and algebras | ✓ | 60 | Probability theory and stochastic processes | ✓ |
| 18 | Category theory; homological algebra | ✓ | 62 | Statistics | ✓ |
| 19 | K-theory | ✓ | 65 | Numerical analysis | ✓ |
| 20 | Group theory and generalizations | ✓ | 68 | Computer science | ✗ |
| 22 | Topological groups, Lie groups | ✓ | 70 | Mechanics of particles and systems | ✗ |
| 26 | Real functions | ✓ | 74 | Mechanics of deformable solids | ✗ |
| 28 | Measure and integration | ✓ | 76 | Fluid mechanics | ✗ |
| 30 | Functions of a complex variable | ✓ | 78 | Optics, electromagnetic theory | ✗ |
| 31 | Potential theory | ✓ | 80 | Classical thermodynamics, heat transfer | ✗ |
| 32 | Several complex variables and analytic spaces | ✓ | 81 | Quantum theory | ✗ |
| 33 | Special functions | ✓ | 82 | Statistical mechanics, structure of matter | ✗ |
| 34 | Ordinary differential equations | ✓ | 83 | Relativity and gravitational theory | ✗ |
| 35 | Partial differential equations | ✓ | 85 | Astronomy and astrophysics | ✗ |
| 37 | Dynamical systems and ergodic theory | ✓ | 86 | Geophysics | ✗ |
| 39 | Difference and functional equations | ✓ | 90 | Operations research, mathematical programming | ✗ |
| 40 | Sequences, series, summability | ✓ | 91 | Game theory, economics, finance... | ✗ |
| 41 | Approximations and expansions | ✓ | 92 | Biology and other natural sciences | ✗ |
| 42 | Harmonic analysis on Euclidean spaces | ✓ | 93 | Systems theory; control | ✗ |
| 43 | Abstract harmonic analysis | ✓ | 94 | Information and communication theory, circuits | ✗ |
| | | | 97 | Mathematics education | ✗ |

*Table 3.* Chosen core MSC2020 topics for our evaluation.

**Sub-Outcome Agreement Rates** We report agreement rates using sub-outcomes rather than outcome categories in Figure 9

# F. Additional Analyses

This appendix reports the four additional analyses summarized in Section 7, conducted after the main experiments to address questions about the protocol's robustness and underlying assumptions.

### F.1. Critic Reliability vs. Answer Correctness

A core premise of CRB is that critiquing is easier than answering: even weak models should be able to reliably identify errors in answers, since spotting a counterexample is structurally simpler than producing a full solution. To test this empirically, we use GPT-5.4-high (which is substantially stronger than all evaluated models) as a ground-truth proxy and measure two quantities per model: *answer correctness rate* (fraction of answers GPT-5.4-high deems correct) and *critique agreement rate* (fraction of critiques on which GPT-5.4-high agrees with the model's verdict).

Table 10 reports the results. Across all eight models, critique agreement rates are uniformly high (0.88–0.96), while answer correctness rates vary significantly (0.16–0.99). This asymmetry holds even for the weakest models in our roster: GPT-3.5 Turbo answers only 15.7% of questions correctly but achieves 95.3% agreement on critiques. The implication is that the bounded verification step at the heart of CRB does not require the verifier to be capable of producing correct answers itself, which is precisely what makes the protocol scale to settings where the evaluator is weaker than the evaluated.

| Human-readable name | Model string | Temp. | Reasoning |
|---|---|---|---|
| GPT-3.5 Turbo | openai/gpt-3.5-turbo-0125 | 0 | N/A |
| GPT-4o | openai/gpt-4o-2024-08-06 | 0 | N/A |
| GPT-5.2 | openai/gpt-5.2-2025-12-11 | 1* | high |
| Phi-4 | microsoft/phi-4 | 0 | high |
| Llama 4 Maverick | meta-llama/llama-4-maverick | 0 | high |
| Claude 4.5 Opus | anthropic/claude-opus-4.5 | 1* | high |
| Gemini 3 Pro (Preview) | google/gemini-3-pro-preview | 1* | high |
| DeepSeek v3.2 Speciale | deepseek/deepseek-v3.2-speciale | 0 | high |

*Table 4.* Model roster used in experiments. * indicates the model does not support setting a non-1 temperature.

| Hyperparameter | Value |
|---|---|
| $\sigma_\alpha$ | 5.755 |
| $\sigma_\beta$ | 4.482 |
| $\sigma_\delta$ | 1.000 |

*Table 5.* Hyperparameters used in our experiments. $\sigma_\delta$ is arbitrarily fixed to 1 and $\sigma_\alpha$ and $\sigma_\beta$ are estimated from data.

### F.2. Mutual Incompatibility of Surviving Answers

If CRB had a substantial false-negative rate (i.e., it admitted incorrect answers as critique-resilient), we would expect to find pairs of surviving answers to the same question that are mutually incompatible. We tested this by classifying pairs of non-proven-wrong answers using GPT-5.4-high. Across 3779 pairs spanning 271 protocol-ready questions, 160 (4.23%) were judged mutually incompatible.

Crucially, incompatibility is concentrated in pairs involving weak models: GPT-3.5 pairs show 14.5–18.3% incompatibility, while frontier-model pairs (Claude 4.5 Opus, DeepSeek V3.2, GPT-5.2, Gemini 3 Pro) show 0.0–0.4%. This is consistent with the framework's design: as the critic pool strengthens, the set of incorrect-but-unfalsifiable answers shrinks. We report the full pairwise breakdown in Table 11.

### F.3. Prompt Sensitivity

To test whether the protocol's rankings are sensitive to specific prompt design choices, we re-ran a subset of the experiments with two alternative prompt variants: *no guidance document* (the rubric in Figure 10 is omitted) and *no self-check knowledge* (the benchmarker is not told that its self-answer will be verified). The ablation uses two benchmarkers (GPT-5.2 and Claude 4.5 Opus), seven answerers/critics (all models except Gemini 3 Pro Preview, which was deprecated at the time of the experiment), and GPT-5.4-high as the sole evaluator (i.e., no human adjudication).

The resulting answerer rankings are highly stable across prompt variants: Kendall $W = 0.952$, mean pairwise Spearman $\rho = 0.929$, and mean pairwise Kendall $\tau = 0.873$. Table 12 reports pairwise correlations with bootstrap 95% CIs (computed in the same fashion as in the main results). The strongest variation appears between the two ablated variants (no guidance vs. no self-check knowledge), while either ablated variant is highly correlated with the standard prompt, suggesting that the protocol's measurement of answering capability is robust to the kinds of prompt-level changes that would meaningfully alter what is being asked.

### F.4. $\alpha$–$\beta$ Correlation

We computed Spearman and Kendall correlations between the answerer Elos (Figure 2) and benchmarker Elos (Figure 6) and obtained $\rho = 0.69$ and $\tau = 0.47$, which represents a positive but not strong correlation. For example, Llama 4 Maverick ranks notably higher as an answerer than as a benchmarker, while GPT-4o shows the opposite pattern.

| Model | Valid | Invalid | Failed |
|---|---|---|---|
| GPT-3.5 Turbo | 8 (18.2%) | 35 (79.5%) | 1 (2.3%) |
| GPT-4o | 37 (84.1%) | 7 (15.9%) | 0 (0.0%) |
| GPT-5.2 | 44 (100.0%) | 0 (0.0%) | 0 (0.0%) |
| Claude 4.5 Opus | 44 (100.0%) | 0 (0.0%) | 0 (0.0%) |
| Gemini 3 Pro Preview | 44 (100.0%) | 0 (0.0%) | 0 (0.0%) |
| Llama 4 Maverick | 35 (79.5%) | 9 (20.5%) | 0 (0.0%) |
| Phi-4 | 15 (34.1%) | 29 (65.9%) | 0 (0.0%) |
| DeepSeek V3.2 Speciale | 44 (100.0%) | 0 (0.0%) | 0 (0.0%) |

*Table 6.* Question outcomes for our experiments. Valid: The model successfully generated a question that withstood critique by itself and other models. Invalid: the model successfully generated a question that withstood critique by itself, but not other models. Failed: the model failed to generate a question that withstood its own critique.

| Model | AIME 2025 | BRUMO 2025 | HMMT Feb 2025 |
|---|---|---|---|
| GPT-3.5 Turbo | $5.8 \pm 4.2$ | $16.7 \pm 6.7$ | $1.7 \pm 2.3$ |
| GPT-4o | $7.5 \pm 4.7$ | $15.8 \pm 6.5$ | $0.8 \pm 1.6$ |
| GPT-5.2 | $94.9 \pm 4.0$ | $90.8 \pm 5.2$ | $98.3 \pm 2.3$ |
| Phi-4 | $20.0 \pm 7.2$ | $23.3 \pm 7.6$ | $3.3 \pm 3.2$ |
| Llama 4 Maverick | $16.7 \pm 6.7$ | $27.5 \pm 8.0$ | $10.0 \pm 5.4$ |
| Claude 4.5 Opus | $89.2 \pm 5.6$ | $92.5 \pm 4.7$ | $81.7 \pm 6.9$ |
| Gemini 3 Pro (Preview) | $95.0 \pm 3.9$ | $96.7 \pm 3.2$ | $97.5 \pm 2.8$ |
| DeepSeek v3.2 Speciale | $95.8 \pm 3.6$ | $95.0 \pm 3.9$ | $97.5 \pm 2.8$ |

*Table 7.* Computed external benchmarks (with CIs) using Matharena.

# G. Hyperparameter Selection via Empirical Bayes

This appendix specifies the objective used to select the prior scales $\sigma = (\sigma_\beta, \sigma_\alpha, \sigma_\delta)$ in the itemized bipartite Bradley-Terry model, using a Laplace (Gaussian) approximation to the marginal likelihood.

### G.1. Model Recap and Notation

Let $b \in \{1, \ldots, B\}$ index *answerer systems* and let questions be indexed as $q_{a,i}$, the $i$-th question authored by system $a \in \{1, \ldots, A\}$. For each eligible attempt where answerer $b$ answers $q_{a,i}$, we record an outcome

$$y_{(a,i),b} \in \{0, 1\},$$

where $y = 1$ denotes an *answerer win* (critique-resilient answer under the protocol) and $y = 0$ denotes a *benchmarker win* (failure to answer, or an upheld critique, including upheld *obscurity*, which counts as a benchmarker win by protocol design).

We exclude from this model any *dropped* episodes (e.g., unresolved adjudication, feasibility-gate failure, upheld ill-posedness invalidating the question, missing artifacts/timeouts).

Define the linear predictor and success probability:

$$\eta_{(a,i),b} := \beta_b - \alpha_a - \delta_{a,i}, \qquad p_{(a,i),b} := \sigma(\eta_{(a,i),b}) = \frac{1}{1 + \exp(-\eta_{(a,i),b})}.$$

Let $\mathcal{D}$ be the set of eligible edges $((a,i), b)$. The Bernoulli likelihood is

$$p(y \mid \phi) = \prod_{((a,i),b) \in \mathcal{D}} p_{(a,i),b}^{y_{(a,i),b}} \left(1 - p_{(a,i),b}\right)^{1 - y_{(a,i),b}},$$

with parameters $\phi := (\beta, \alpha, \delta)$.

We place independent zero-mean Gaussian priors:

$$\beta_b \sim \mathcal{N}(0, \sigma_\beta^2), \qquad \alpha_a \sim \mathcal{N}(0, \sigma_\alpha^2), \qquad \delta_{a,i} \sim \mathcal{N}(0, \sigma_\delta^2).$$

| Model | Spearman $\rho$ | Kendall $\tau$ |
|---|---|---|
| GPT-3.5 Turbo | 1.000 | 1.000 |
| GPT-4o | 1.000 | 1.000 |
| GPT-5.2 | 1.000 | 1.000 |
| Claude 4.5 Opus | 1.000 | 1.000 |
| Gemini 3 Pro Preview | 1.000 | 1.000 |
| Llama 4 Maverick | 1.000 | 1.000 |
| Phi-4 | 1.000 | 1.000 |
| DeepSeek V3.2 Speciale | 1.000 | 1.000 |

*Table 8.* Benchmarker Spearman $\rho$ and Kendall $\tau$ between answer Elos computed with LLM adjudications and Elos computed with human adjudication.

| Model | Spearman $\rho$ | Kendall $\tau$ |
|---|---|---|
| GPT-3.5 Turbo | 1.000 | 1.000 |
| GPT-4o | 0.976 | 0.929 |
| GPT-5.2 | 1.000 | 1.000 |
| Claude 4.5 Opus | 1.000 | 1.000 |
| Gemini 3 Pro Preview | 0.976 | 0.929 |
| Llama 4 Maverick | 0.976 | 0.929 |
| Phi-4 | 0.976 | 0.929 |
| DeepSeek V3.2 Speciale | 0.976 | 0.929 |

*Table 9.* Answerer Spearman $\rho$ and Kendall $\tau$ between answer Elos computed with LLM adjudications and Elos computed with human adjudication.

## G.2. Identifiability constraints

The likelihood depends on $\phi$ only through differences $\beta_b - \alpha_a - \delta_{a,i}$, so it is invariant to a global shift $(\beta, \alpha, \delta) \mapsto (\beta + c, \alpha + c, \delta)$. We remove this degree of freedom via a centering constraint, applied consistently across the main fit and all resampling refits:

$$\sum_{b=1}^{B} \beta_b = 0.$$

In practice, we enforce this by reparameterization (dropping one degree of freedom).

## G.3. MAP estimate (penalized logistic regression)

For fixed $\sigma$, the MAP estimate maximizes the log joint:

$$\hat{\phi}(\sigma) = \arg \max_{\phi} \left\{ \log p(y \mid \phi) + \log p(\phi \mid \sigma) \right\},$$

subject to the identifiability constraint.

Writing the objective explicitly yields L2-regularized logistic regression:

$$\log p(y \mid \phi) = \sum_{((a,i),b) \in \mathcal{D}} \left[ y_{(a,i),b} \log p_{(a,i),b} + (1 - y_{(a,i),b}) \log(1 - p_{(a,i),b}) \right],$$

$$\log p(\phi \mid \sigma) = -\frac{1}{2\sigma_\beta^2} \sum_{b=1}^{B} \beta_b^2 - \frac{1}{2\sigma_\alpha^2} \sum_{a=1}^{A} \alpha_a^2 - \frac{1}{2\sigma_\delta^2} \sum_{a,i} \delta_{a,i}^2 + C(\sigma),$$

where $C(\sigma)$ collects normalization terms.

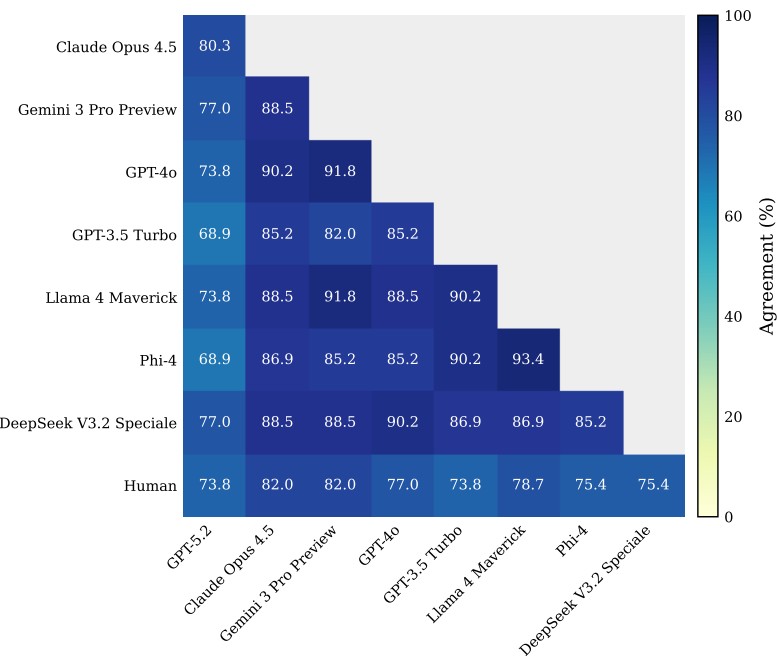

*Figure 9.* Agreement rates between human evaluators and models, when using sub-outcomes rather than outcomes.

| Model | Answer Correctness | Critique Agreement | Critique F1 |
|---|---|---|---|
| Claude 4.5 Opus | 0.967 | 0.946 | 0.972 |
| DeepSeek V3.2 Speciale | 0.989 | 0.939 | 0.968 |
| Gemini 3 Pro Preview | 0.970 | 0.963 | 0.981 |
| GPT-5.2 | 0.974 | 0.944 | 0.970 |
| Llama 4 Maverick | 0.445 | 0.919 | 0.957 |
| GPT-4o | 0.391 | 0.883 | 0.938 |
| Phi-4 | 0.315 | 0.924 | 0.961 |
| GPT-3.5 Turbo | 0.157 | 0.953 | 0.976 |

*Table 10.* Answer correctness and critique agreement rates per model, using GPT-5.4-high as ground-truth proxy. Critique agreement rates are uniformly high across all models, while answer correctness rates vary widely.

## G.4. Empirical Bayes Objective

We select $\sigma$ by maximizing the marginal likelihood:

$$\hat{\sigma} = \arg\max_{\sigma} \log p(y \mid \sigma), \qquad p(y \mid \sigma) = \int p(y \mid \phi)\, p(\phi \mid \sigma)\, d\phi.$$

This trades off fit and flexibility via an "Occam" penalty induced by integrating out $\phi$.

## G.5. Laplace Approximation Around the MAP

Let

$$\mathcal{L}(\phi; \sigma) := \log p(y \mid \phi) + \log p(\phi \mid \sigma),$$

and let $\hat{\phi}(\sigma)$ be the constrained MAP solution. Let $H(\sigma)$ be the negative Hessian of $\mathcal{L}$ at $\hat{\phi}(\sigma)$, computed on the constrained (reparameterized) space:

$$H(\sigma) := -\nabla_{\phi}^2 \mathcal{L}(\phi; \sigma)\big|_{\phi=\hat{\phi}(\sigma)}.$$

Then the Laplace approximation is

$$\log p(y \mid \sigma) \approx \mathcal{L}(\hat{\phi}(\sigma); \sigma) + \frac{k}{2}\log(2\pi) - \frac{1}{2}\log|H(\sigma)|,$$

| Model A | Model B | Incompatible / Total | Rate |
|---|---|---|---|
| GPT-3.5 Turbo | GPT-4o | 11 / 60 | 18.3% |
| Phi-4 | GPT-3.5 Turbo | 10 / 59 | 16.9% |
| Claude 4.5 Opus | GPT-3.5 Turbo | 9 / 62 | 14.5% |
| DeepSeek V3.2 Speciale | GPT-3.5 Turbo | 9 / 62 | 14.5% |
| Gemini 3 Pro Preview | GPT-3.5 Turbo | 9 / 62 | 14.5% |
| Llama 4 Maverick | GPT-3.5 Turbo | 9 / 62 | 14.5% |
| GPT-3.5 Turbo | GPT-5.2 | 9 / 62 | 14.5% |
| Llama 4 Maverick | Phi-4 | 8 / 102 | 7.8% |
| DeepSeek V3.2 Speciale | Llama 4 Maverick | 10 / 152 | 6.6% |
| Gemini 3 Pro Preview | Llama 4 Maverick | 10 / 152 | 6.6% |
| Llama 4 Maverick | GPT-5.2 | 10 / 152 | 6.6% |
| Claude 4.5 Opus | Llama 4 Maverick | 9 / 151 | 6.0% |
| DeepSeek V3.2 Speciale | Phi-4 | 6 / 106 | 5.7% |
| Llama 4 Maverick | GPT-4o | 6 / 106 | 5.7% |
| Claude 4.5 Opus | Phi-4 | 6 / 107 | 5.6% |
| Gemini 3 Pro Preview | Phi-4 | 6 / 107 | 5.6% |
| Phi-4 | GPT-5.2 | 6 / 107 | 5.6% |
| Phi-4 | GPT-4o | 5 / 94 | 5.3% |
| Gemini 3 Pro Preview | GPT-4o | 3 / 111 | 2.7% |
| Claude 4.5 Opus | GPT-4o | 2 / 111 | 1.8% |
| DeepSeek V3.2 Speciale | GPT-4o | 2 / 111 | 1.8% |
| GPT-4o | GPT-5.2 | 2 / 111 | 1.8% |
| Claude 4.5 Opus | Gemini 3 Pro Preview | 1 / 259 | 0.4% |
| DeepSeek V3.2 Speciale | Gemini 3 Pro Preview | 1 / 260 | 0.4% |
| Gemini 3 Pro Preview | GPT-5.2 | 1 / 262 | 0.4% |
| DeepSeek V3.2 Speciale | GPT-5.2 | 0 / 265 | 0.0% |
| Claude 4.5 Opus | GPT-5.2 | 0 / 263 | 0.0% |
| Claude 4.5 Opus | DeepSeek V3.2 Speciale | 0 / 261 | 0.0% |

*Table 11.* Pairwise mutual incompatibility rates among non-proven-wrong answer pairs, classified by GPT-5.4-high. Pairs involving GPT-3.5 show the highest rates; pairs among the four strongest models show near-zero rates.

| Prompt Pair | Spearman $\rho$ | Kendall $\tau$ |
|---|---|---|
| No guidance vs. no self-check | 0.893 [0.821, 0.964] | 0.810 [0.619, 0.905] |
| No guidance vs. standard | 1.000 [0.857, 1.000] | 1.000 [0.714, 1.000] |
| No self-check vs. standard | 0.893 [0.857, 1.000] | 0.810 [0.714, 1.000] |

*Table 12.* Pairwise correlations between answerer rankings under three prompt variants, with bootstrap 95% CIs.

where $k$ is the number of free parameters after enforcing identifiability.

We optimize this approximation over $\log \sigma$ for numerical stability.

### G.6. Practical Computation and Selection Protocol

- **Selection protocol:** we use a coarse grid search over $\log \sigma$ (or over each $\log \sigma_\bullet$) followed by local refinement.

- **Log-determinant:** $H(\sigma)$ is sparse; $\log |H|$ is computed via sparse Cholesky factorization $H = LL^\top \Rightarrow \log |H| = 2 \sum_j \log L_{jj}$.

## H. Uncertainty via Bootstrap

This appendix describes the bootstrap procedure used to quantify uncertainty in $\beta_b$ and $\alpha_a$.

### H.1. Cluster Bootstrap over Questions

Because all outcomes for the same question instance $q_{a,i}$ share $\delta_{a,i}$, resampling edges independently is inappropriate. Let

$$\mathcal{Q} := \{(a, i)\}$$

```
# Task: Generate a Challenging Mathematics Problem

You are acting as a **tester** in a benchmarking framework. Your goal is to create a single, challenging but solvable mathematics
    problem along with a complete, verifiable solution.

## Quality Rubric

{{guidance_text}}

## Topic

Generate a problem in the following domain: **{{topic}}**

{{previous_context}}
{{previous_questions}}

## Output Format

Use exactly this structure:

[QUESTION]
<Your problem statement here>

[ANSWER]
<Your complete solution here>

**Important**: Your solution will be verified first (self-solve gate). If it fails verification, the question will be rejected without
    being used to test other models.
Use standard LaTeX notation for mathematical expressions where appropriate, delineated with $ or $$.
```

*Figure 10.* Question prompt. {{previous_context}} and {{previous_questions}} are only included when there are prior attempts.

be the set of question instances appearing in $\mathcal{D}$. For bootstrap replicate $t = 1, \ldots, T$:

1. Sample $|\mathcal{Q}|$ question instances with replacement to form a multiset $\mathcal{Q}^{(t)}$.

2. Construct $\mathcal{D}^{(t)}$ by including **all** edges for each sampled $(a, i)$, with multiplicity equal to its resample count.

3. Refit the MAP model on $\mathcal{D}^{(t)}$ using the **fixed** hyperparameters $\hat{\sigma}$ selected in Appendix G, and the same identifiability convention.

4. Record $\hat{\beta}_b^{(t)}$ and $\hat{\alpha}_a^{(t)}$.

We report percentile confidence intervals (2.5%–97.5%) and bootstrap standard errors.

### H.2. Elo-Like Presentation

For readability, strengths are mapped to an Elo-like scale via an affine transform:

$$\mathrm{Elo}(\beta_b) = c + k\beta_b, \qquad \mathrm{Elo}(\alpha_a) = c' + k'\alpha_a.$$

This does not affect ordering. A conventional choice is $k = 400/\ln 10$, so a difference of 400 corresponds to 10:1 odds under the logistic link; we treat this as a presentation convenience only.

and all bootstrap refits.

```
# Task: Solve the Mathematics Question

You are acting as a **testee** in a benchmarking framework. Provide a complete, rigorous answer to the question below.

## Answer Quality Requirements

{{guidance_text}}

## The Question

{{question}}

## Your Response

Provide your complete answer below. Use standard LaTeX notation for mathematical expressions where appropriate, delineated with $ or $$.

If the question is ill-posed, explicitly state this and explain why rather than attempting to answer.
```

*Figure 11.* Answer prompt.

```
# Task: Evaluate Your Own Answer

Review the answer you provided to the question below and assess whether it meets the quality standards.

## Question

{{question}}

## Your Answer

{{answer}}

## Answer Quality Requirements

{{answer_guidance}}

## Evaluation Rubric

{{self_critique_guidance}}

## Required Output Format

Return ONLY a JSON object that matches this JSON Schema (no additional text):

```json
{
  "type": "object",
  "additionalProperties": false,
  "required": ["verdict", "ill_posed", "issues", "improvements"],
  "properties": {
    "verdict": { "type": "string", "enum": ["pass", "fail"] },
    "ill_posed": { "type": "boolean" },
    "issues": { "type": "array", "items": { "type": "string" } },
    "improvements": { "type": "string" }
  }
}
```

**Field Descriptions:**
- `verdict`: "pass" if the answer is correct and complete, "fail" otherwise
- `ill_posed`: true if the question itself is unanswerable as stated, false otherwise
- `issues`: List of specific problems with the answer (empty list if none)
- `improvements`: Short, concrete guidance on how to fix the answer (empty string if verdict is "pass")

Use standard LaTeX notation for mathematical expressions where appropriate, delineated with $ or $$.
```
```

*Figure 12.* Self-check prompt.

```
# Task: Improve Your Answer

Revise your answer to address the issues identified in the feedback below.

## Question

{{question}}

## Current Answer

{{answer}}

## Feedback

{{feedback}}

## Quality Standards

{{guidance_text}}

## Your Revised Answer

Provide only the improved answer below (no meta-commentary). Use standard LaTeX notation for mathematical expressions where appropriate,
      delineated with $ or $$.
```

*Figure 13.* Answer refinement prompt.

```
# Task: Verify Answer Correctness

You are evaluating an answer provided by a model. Your role is **verification**, not independent solution generation.

## Critique Guidelines

{{critique_guidance}}

## Question

{{question}}

## Answer to Evaluate

{{answer}}

## Required Output Format

Return ONLY a JSON object that matches this JSON Schema (no additional text):

```json
{
  "type": "object",
  "additionalProperties": false,
  "required": ["verdict", "notes"],
  "properties": {
    "verdict": {
      "type": "string",
      "enum": ["correct", "incorrect", "insufficient", "obscure"]
    },
    "notes": { "type": "string" },
    "suggestions": { "type": "string" }
  }
}
```

**Verdict Definitions:**
- `"correct"`: Mathematically sound, complete, and adequately rigorous
- `"incorrect"`: Contains errors, invalid reasoning, or fundamental flaws
- `"insufficient"`: Partially correct but meaningfully incomplete
- `"obscure"`: Too unclear to verify

Use standard LaTeX notation for mathematical expressions where appropriate, delineated with $ or $$.
```

*Figure 14.* Critique prompt.

```
# Task: Review Your Critique for Accuracy

Assess whether your critique correctly identifies issues (if any) and provides accurate, evidence-based reasoning.
Also verify that your critique follows the required critique JSON format.

## Question

{{question}}

## Answer Being Critiqued

{{answer}}

## Your Critique

{{critique}}

## Evaluation Standards

{{critique_guidance}}

## Critique Format Check

Your critique must be valid JSON with `verdict` in {correct, incorrect, insufficient, obscure}, `notes` as a string, and optional `
      suggestions`. If the format is invalid, set verdict to "fail" and list the formatting problems in `issues`.

## Required Output Format

Return ONLY a JSON object that matches this JSON Schema (no additional text):

```json
{
  "type": "object",
  "additionalProperties": false,
  "required": ["verdict", "issues", "improvements"],
  "properties": {
    "verdict": { "type": "string", "enum": ["pass", "fail"] },
    "issues": { "type": "array", "items": { "type": "string" } },
    "improvements": { "type": "string" }
  }
}
```

- `verdict`: "pass" if your critique is accurate and well-justified, "fail" if it needs revision
- `issues`: Specific problems with your critique (e.g., incorrect claims, missing evidence)
- `improvements`: Guidance on how to make the critique more accurate

Use standard LaTeX notation for mathematical expressions where appropriate, delineated with $ or $$.
```

*Figure 15.* Critique self-check prompt.

```
# Task: Improve Your Critique

Revise your critique to address the issues identified below.

## Question

{{question}}

## Answer Being Critiqued

{{answer}}

## Your Current Critique

{{critique}}

## Feedback on Your Critique

{{feedback}}

## Required Output Format

Return ONLY a JSON object that matches this JSON Schema (no additional text):

```json
{
  "type": "object",
  "additionalProperties": false,
  "required": ["verdict", "notes"],
  "properties": {
    "verdict": {
      "type": "string",
      "enum": ["correct", "incorrect", "insufficient", "obscure"]
    },
    "notes": { "type": "string" },
    "suggestions": { "type": "string" }
  }
}
```

**Verdict Definitions:**
- `"correct"`: Mathematically sound, complete, and adequately rigorous
- `"incorrect"`: Contains errors, invalid reasoning, or fundamental flaws
- `"insufficient"`: Partially correct but meaningfully incomplete
- `"obscure"`: Too unclear to verify

Provide your improved critique using the JSON format above.
Use standard LaTeX notation for mathematical expressions where appropriate, delineated with $ or $$.
```

*Figure 16.* Critique refinement prompt.

```
You are a neutral judge. Your task: Determine if ALICE'S CLAIM is valid.
Alice claims the question is ill-posed. Bob defends that it is well-posed.
The debate helps you assess the claim. If Bob concedes or acknowledges the issue, this CONFIRMS Alice was right.
Use the guidance below and ignore any party identity cues.

Judgment guidance:
{{guidance_j}}

Question quality guidance:
{{guidance_q}}

Question:
{{question}}

Answer (context only):
{{answer}}

Debate transcript:
{{debate}}

Return ONLY a JSON object with this schema:
{
  "verdict": "claimant_wins" | "defender_wins_incorrect" | "wrong_problem" | "mixed" | "unknown",
  "confidence": 1-5,
  "reasoning": "<concise justification>"
}

Verdict meanings:
- "claimant_wins": Alice's claim is correct - the question is ill-posed (use this even if Bob concedes)
- "defender_wins_incorrect": Alice's claim is incorrect - Bob shows the question is well-posed and Alice misidentified the issue
- "wrong_problem": The question has issues, but Alice's specific diagnosis is incorrect
- "mixed": Alice makes multiple claims, some correct and some incorrect
- "unknown": Cannot determine if Alice's claim is valid with confidence

Confidence scale: 1=very uncertain, 2=uncertain, 3=moderate, 4=confident, 5=very confident
```

*Figure 17.* Automated evaluation prompt (ill-posedness claims).

```
You are a neutral judge. Your task: Determine if ALICE'S CRITIQUE is correct.
Alice claims Bob's answer has errors/issues. Bob defends the answer.
The debate helps you assess the critique. If Bob concedes, fixes the issue, or provides missing justification, this CONFIRMS Alice was
     right.
Use the guidance below and ignore any party identity cues.

Judgment guidance:
{{guidance_j}}

Answer quality guidance:
{{guidance_a}}

Critique guidance:
{{guidance_c}}

Question:
{{question}}

Answer:
{{answer}}

Critique:
{{critique}}

Debate transcript:
{{debate}}

Return ONLY a JSON object with this schema:
{
  "verdict": "claimant_wins" | "defender_wins_incorrect" | "defender_wins_minor" | "wrong_problem" | "mixed" | "unknown",
  "confidence": 1-5,
  "reasoning": "<concise justification>"
}

Verdict meanings:
- "claimant_wins": Alice's critique is correct - the answer has substantive flaws (use this even if Bob concedes/fixes)
- "defender_wins_incorrect": Alice's critique is incorrect - Bob shows the answer is correct and Alice misidentified a problem
- "defender_wins_minor": Alice's critique is technically correct but about very minor (stylistic only) issues
- "wrong_problem": There are issues with the answer, but Alice's specific diagnosis is incorrect
- "mixed": Alice makes multiple claims, some correct and some incorrect
- "unknown": Cannot determine if Alice's critique is valid with confidence

Confidence scale: 1=very uncertain, 2=uncertain, 3=moderate, 4=confident, 5=very confident
```

*Figure 18.* Automated evaluation prompt (incorrectness claims).

