# OpenReview forum: "Benchmarking at the Edge of Comprehension"
_ICML.cc/2026/Conference — ICML 2026 spotlight_

### Official Review · Reviewer_aQbn · 2026-03-11

**Soundness:** 4
**Presentation:** 4
**Significance:** 3
**Originality:** 3
**Overall Recommendation:** 5
**Confidence:** 4

**Summary:**

The paper proposes a new protocol for benchmarking LLMs that overcomes issues with the quick saturation of maths benchmarks we've seen due to quick progress on LLM capability. Instead of making hard tasks and checking the output, the authors suggest to accept that soon humans will be incapable of generating challenging enough tasks or wouldn't be able to manually check a full response.

Instead the paper suggests to focus on what they call 'witness admitting domains' where an answer must be fully correct to be correct and thus can be attacked in parts. This assumes that picking and attacking a part of e.g. a proof would be easier than verifying the full proof. That attack would still be automatic but the evidence of a local incorrectness could be potentially verified by humans. The overall benchmark moves from various proxies of 'answer is correct' towards 'cannot prove answer is [partially] wrong'.

The paper proses a protocol where 2 models A and B are evaluated against each other: A makes a (math) question, B gives an answer, A tries to find a flaw and presents it to a judge. If the answer is indeed flawed A gets a point for being clever.

Model scores are computed by a Bradley-Terry model that accounts for answerer-strength and benchmarker strength.

The authors convincingly answer two research questions: First, they show that the model ratings they infer with their protocol correlate well with model performance on current benchmarks. Importantly they do not need a benchmark dataset for their evaluation, the only input to the tournaments are topics like 'Statistics', the rest is generated by the contestants. Second, they show that using weak models as judges to verify possible flaws leads to a very similar outcome, giving us hope that when humans become the weak models, last-generation models can still be reasonable judges.

**Compliance With Llm Reviewing Policy:**

Affirmed.

**Final Justification:**

I only had some minor issues and questions that were fully addressed in the rebuttal. The paper can be published.

**Key Questions For Authors:**

(1): Evaluation game: In Definition 3.4 it says that B wins if [...] no valid ill-posedness critique is upheld. From section 3.1 I understood that the second claim type would be a critique of the question, possibly done by B, and if upheld lead to a win for B. Maybe rephrase to:

B wins if either an ill-posedness critique for q is upheld or a_B is critique resilient: no incorrectness or obscurity claim is upheld.

(1.1) Feasibility Gating: Similar to the issue above, In Section 4.1, wouldn't the dropping of the question not count as a win for B?

(2): Separation: I don't see the need to use the same model for question generation and as a critic, even though that obviously helps with the two-sides protocol. Would it make sense to disentangle these?

(3) Analysis of alpha scores. I understand that the main results are the better scores but you say yourselves that looking at alpha would be helpful to understand experimental results. Could these be included in one of the plots?

**Limitations:**

yes

**Strengths And Weaknesses:**

Soundness: Good, the paper is easy to follow and I wasn't able to identify any fundamental problems that cannot easily be fixed. I like the claim types in Section 3, this makes sure that verification goes beyond 'all the steps look reasonable'.

Presentation is great, good writing, great introduction, motivation and related work. Good explanation of their proposed protocol and experiments. Good structure and helpful appendix.

Significance: Good, as we a quickly heading towards AI that does very well on the math type tasks they address in this paper, manually generating benchmarks and verifying answers will soon meet a dead end, so the proposed protocol and rating scheme is important.

One weakness comes from resolution. It seems that with the 352 questions they generated, it is easily possible to tell the top models from some weaker ones, but within the clusters differences are not significant, making it hard to track smaller improvements. This might go away with running more tournaments.

Another weakness that is possibly unavoidable is that new models might come in and invalidate earlier results because the new models might find flaws in previously accepted answers, which the authors acknowledge.

Originality: Medium - Much of this work is a logical expansion of the LLM as Judge, Scalable Oversight, Rubrics Based Eval, Critique Model etc work, which I think is well explained in Section 8. That being said, for the set of problems where this protocol is applicable

---

> ### Author Rebuttal · Authors · 2026-03-30
>
> Dear Reviewer aQbn,
>
> Thank you for your careful and constructive review. We appreciate your positive assessment of the protocol's soundness, the presentation, and the significance of the problem. We address each of your questions below.
>
> **1 & 1.1. Definition 3.4 and Feasibility Gating**
>
> Thank you for catching this! The phrasing in Definition 3.4 is indeed incorrect as written. The intended semantics (which were used in the experiments) are:
>
> * B wins if $a_B$ is critique-resilient: no valid incorrectness or obscurity critique is upheld.
> * A wins if B fails to answer, or if a valid critique establishes that $a_B$ is incorrect or incomplete.
> * The outcome is dropped if the protocol cannot resolve the relevant claims within its adjudication budget, or if an ill-posedness critique is upheld.
>
> In other words, upheld ill-posedness leads to a drop, not a win for either party. The reason for this is mostly statistical: in the bipartite Bradley-Terry model, the difficulty parameter $\theta_{a,i}$ for an invalid question has no meaningful interpretation, so including it would contaminate the latent space. We will correct the phrasing for the camera-ready version.
>
> **2. Disentangling Benchmarker and Critic**
>
> This is a great question, and one we spent considerable time on during the design phase. We originally developed a tripartite model matching your intuition (Questioner, Answerer, Checker), but had to abandon it due to a fundamental problem: there is no epistemically valid way to distinguish "B wins because the question is easy" (which should penalize the Questioner, reward the Answerer, and leave the Checker unchanged) from "B wins because no mistake could be spotted" (which should leave the Questioner unchanged, reward the Answerer, and penalize the Checker). Since these two explanations produce the same observable outcome, we don’t have the information required to perform the three-way decomposition. We solved this by merging Questioner and Checker into a single Benchmarker role.
>
> Note: there are edge cases where partial disentanglement is possible (e.g., if GPT-5.2 spots an error that GPT-3.5 misses, this is informative about GPT-3.5’s critic capability), but these cannot be exploited consistently enough to support a full tripartite model. We will add a discussion of this design decision and the identifiability argument to the camera-ready version, as we believe it may be of interest to readers considering extensions of our framework.
>
> **3. Analysis of $\alpha$ Scores**
>
> We report benchmarker Elos with 95% CIs in Figure 5 (Appendix C, page 17). We agree these deserve more visibility and will consider promoting them to the main text in the camera-ready version, space permitting.
>
> We also computed correlations and found that alphas and betas are moderately, but not fully, correlated (Spearman $\rho$ = 0.69, Kendall $\tau$ = 0.47), which is interesting and shows that certain models might be good benchmarkers but not good answerers (and vice versa).
>
> **On Resolution**
>
> We agree that the current scale separates tiers clearly but does not resolve within-cluster differences. We expect that running at larger scale would shrink the confidence intervals and reveal finer-grained separation, with the notable advantage that, unlike traditional benchmarks, scaling CRB requires no additional human-authored problems but only compute and the occasional human adjudication.
>
> **On Invalidation**
>
> You rightly note that stronger future models could identify errors in previously accepted answers. This is correct and worth emphasizing: critique-resilient correctness is relative to the critic class by construction (Definition 3.3), so the scores should always be read as "capability against the best available critics at evaluation time." The upside is that re-evaluation is mechanically straightforward: one re-runs the critique stage with the new critic and updates the Bradley-Terry fit, which requires no new human-authored questions (only the occasional adjudication). We will make this interpretation more explicit in the camera-ready version.
>
> We hope these clarifications address your concerns. Let us know if there are further questions or additional analyses that would increase your confidence in the paper!
>
> Thank you again,
>
> The Authors

---

> > ### Author Rebuttal · Reviewer_aQbn · 2026-04-02
> >
> > Thanks a lot your clarifications, in particualr regarding the infeasibility of disentanglement!
> >
> > I carefully reviewed your response and the other reviews and still think that 5 is the appropriate rating and that this paper should be accepted.

---

> > > ### Author Response · Authors · 2026-04-04
> > >
> > > Thank you!
> > >
> > > The Authors

---

### Official Review · Reviewer_nEi7 · 2026-03-13

**Soundness:** 3
**Presentation:** 3
**Significance:** 3
**Originality:** 3
**Overall Recommendation:** 5
**Confidence:** 4

**Summary:**

This paper studies the problem of benchmarking frontier models that have capabilities that are difficult or impractical for humans to directly evaluate. The main approach is to consider problems where there is a short witness demonstrating that a purported solution is false. For example, a mistake in the proof of a single lemma for a 50 page AI-generated mathematical proof. In this case, a human might have trouble checking the full proof, but if a specific lemma is pointed out with a flaw, it would be easier for the human to verify this. Given this type of problem, the authors design a game between a *benchmarker* and an *answerer*. The benchmarker proposes a question and the answerer answers it. After this, the benchmarker gives a critique consisting of a short witness that is supposed to invalidate the answerer's solution. There is also an initial feasibility-gating step for each question, where the benchmarker must demonstrate that they can produce an answer to their own question that can survive critique. At the end of this game, the benchmarker wins if they produce a feasible question which the answerer fails to provide a solution that survives critique. If the solution does survive, then the answerer wins. The witnesses proposed in critiques are judged by a set of LLMs, where if the judgement is not unanimous, then the witness is escalated to human judges.

This game gives pairwise win rates for every LLM involved. The authors fit a modified Bradley-Terry model to this matrix of pairwise win-rates. The model parameters give scores for answer-capability, benchmarker-capability, as well as individual question-difficulty. The authors run experiments in the mathematical domain with several frontier LLMs, as well as with older and smaller models. The results demonstrate that their method produces scores consistent with other metrics of mathematical capability of LLMs. They also demonstrate that their protocol makes the problem easier for weaker judges, as LLM judgements even from smaller models have very high correlation with human judgements.

**Compliance With Llm Reviewing Policy:**

Affirmed.

**Final Justification:**

Reading the rebuttal and the other reviews has reinforced my prior assessment that this is an interesting and novel approach to benchmarking. I maintain that this paper should be accepted.

**Key Questions For Authors:**

Do you have any sense of why the benchmarker scores reported in the appendix allow us to distinguish the different frontier models to some degree, but the question-answerer scores in the main body do not? Could this be an issue with eliciting benchmarker capability or is it possibly some other feature of the pairwise setup?

**Limitations:**

yes

**Strengths And Weaknesses:**

**Strengths**

The paper addresses a critical problem: the difficulty of obtaining ground-truth labels for questions that challenge frontier models. The paper proposes a method that allows failures in complex solutions to be efficiently verified by leveraging the power of LLMs to point out flaws that are easier for human judges to verify. The experimental results correspond well with this conceptual motivation.

The paper is well-written and easy to follow, and the rigorous statistical techniques deployed in fitting the Bradley-Terry model make the results seem very solid.

**Weaknesses**

One possible weakness of the empirical results is that it is unclear that the models' capabilities have been fully elicited in the benchmarker role. Certainly all of the frontier models involved have undergone extensive training for answering mathematical questions, and so it seems plausible that they are doing about as well as they can. However, the problem of generating difficult-yet-feasible questions is likely not seen much in training, and so it might have made sense to do some sort of best-of-n sampling to select for the hardest feasible problems the model could generate. I recognize that best-of-n was already used to get the models to generate just one feasible problem, so I understand that additional sampling may have been too costly. Still, it might improve the discriminative abilities of the benchmark to try to elicit harder problems form the benchmarker models. As it stands, the benchmark doesn't seem sensitive enough to distinguish the mathematical capabilities of frontier models.

I believe that the related work section already does a good job of covering a wide variety of background on this topic. However, a minor complaint is how the paper poses its approach in relation to scalable oversight. The main approach, i.e. that one can use an interaction between powerful AIs to allow human judges to more efficiently verify solutions to problems, is precisely the definition of scalable oversight. In addition, the question of exactly when bounded verification of localized claims is possible has been studied theoretically in [1]. This theoretical work provides precisely the formal mathematical definition discussed in footnote 2 for the informal Definition 2.1. This theory actually makes the results of this submission stronger, because it rigorously shows that a much broader variety of problems admit the sort of verification necessary for the proposed benchmarking approach to work.

[1] Brown-Cohen, Irving, and Piliouras. Scalable AI safety via doubly-efficient debate. In Proceedings of the 41st International Conference on Machine Learning (ICML'24)

---

> ### Author Rebuttal · Authors · 2026-03-30
>
> Dear Reviewer nEi7,
>
> Thank you for your thoughtful review. We appreciate your recognition of the core contributions and your positive assessment of the statistical methodology, as well as the strong score.
> We address each point below.
>
> **Elicitation of Benchmarker Capability**
>
> We agree that benchmarking capability can be elicited further beyond our baseline. Besides best-of-n sampling, another interesting approach would be a binary-search-inspired strategy that adaptively tunes difficulty based on whether the benchmarker can self-answer. We expect these would yield harder, more discriminative questions at modest additional cost.
> An appealing property of CRB is that such improvements are purely at the protocol level: they sharpen the measurement without changing the underlying framework or what is being measured. Since the answerer scores $\beta$ are the primary output, stronger benchmarkers translate directly into better statistical power to separate answerers.
>
> **Relationship to Brown-Cohen et al.**
>
> Thank you for pointing us to that paper\! This is an excellent reference that we should have cited. Their framework provides a very useful lens on the kind of bounded local verification captured by our Definition 2.1 (with the appropriate adaptations, of course). Also, their results strengthen our argument that evaluation can remain viable even when full end-to-end verification is infeasible. We will cite it in the camera-ready version and integrate it into the discussion of Section 2.2 and the related work.
>
> More broadly, we agree that our framework is closely related to scalable oversight, and we will make that connection more explicit. The distinction we would draw is that the scalable oversight literature typically studies whether debate can enable the oversight of stronger systems, while CRB uses the same structure for evaluation.
>
> **Why Benchmarker Elos Discriminate Better Than Answerer Elos**
>
> Our intuition is that answering capability is heavily optimized for by model deployers, and as a consequence frontier models converge to similar performance. This pattern is also reflected in traditional benchmarks (Table 7), where the top four models have similar scores on AIME, BRUMO, and HMMT. By contrast, question generation is not a standard training objective, so there is more natural variability between models on this axis. That said, we stress that the answerer scores $\beta$ are the more meaningful output of CRB: $\alpha$ is useful for interpreting the experiments but should not be treated as an intrinsic capability measure outside the protocol (as noted in Section 5.3).
>
> We hope that these clarifications address your concerns, and thanks again for the Brown-Cohen reference (which will definitely strengthen the paper). Let us know if you have any questions, or if there are any experiments that would increase your confidence in the paper’s quality.
>
> Thank you again for the review!
>
> The Authors

---

> > ### Author Rebuttal · Reviewer_nEi7 · 2026-04-02
> >
> > Thanks for your response and explanations. I like the idea of binary search for the hardest possible question that the benchmarker can themselves answer. Overall this is an interesting and novel approach to benchmarking and I recommend acceptance.

---

> > > ### Author Response · Authors · 2026-04-02
> > >
> > > Thank you!
> > >
> > > The Authors

---

### Official Review · Reviewer_Uovv · 2026-03-17

**Soundness:** 2
**Presentation:** 3
**Significance:** 3
**Originality:** 3
**Overall Recommendation:** 5
**Confidence:** 4

**Summary:**

This paper proposes a new benchmarking paradigm in which competing models both propose and answer benchmark questions. Models must propose problems that they themselves can answer, and are rewarded when other models fail to answer them. When an answer is generated by one model, the other (question generating) model has an opportunity to identify flaws in the answer. The key advantage of this approach is that human oversight is not required to check the entire answer (which could be a long mathematical proof), but only to check the local flaws identified during this later verification process. Finally, using a Bradley-Terry Model, one can then extract ELO scores and rankings of question answering ability from these competitions. The paper instantiates this paradigm for math problem solving, successfully extracting reasonable ELO scores for current state-of-the-art models.

**Compliance With Llm Reviewing Policy:**

Affirmed.

**Final Justification:**

The rebuttal addressed my main concerns regarding the experiments and clarified all of my questions. I am happy to recommend acceptance for this paper.

**Key Questions For Authors:**

My primary concern with the paper is with the experimental section, as described above, and the response to that would be the primary factor in any choice to raise my score. That said, I think the following questions are also important to discuss:

Q1: The paper frames "answerer strength" and "benchmarker strength" as the two main capabilities of a model, and they are implicitly assumed to be independent (based on the independent prior assigned in the BT model). However, in my opinion there are really three atomic capabilities: (1) question writing, (2) answering, and (3) error identification. What is called "benchmarker strength" in the paper is really a combination of all three--a good benchmarker clearly has to do both (1) and (3), and it also must be competent at (2) or else its questions will be filtered out by the feasibility filter. It also seems possible that a model could have different abilities to perform any one of these three. What is your opinion of this decomposition as opposed to the one used in the paper?

Q2: I would argue somewhat with the use of the term "comprehension" when describing the "post-comprehension" regime, and similarly with the phrasing of A2 where it says "humans cannot verify" a proposed answer. This suggests that whatever the LLMs are doing is completely inscrutable to human readers. In reality, proofs generated by LLMs should in principle be comprehendible by an expert human reader, but it would be impractically laborious for the purposes of benchmarking. I think that in order to have requirement R2, only this weaker form of A2 (which says that human verification is impractical, not impossible) suffices. Do you agree with this?

Q3: Could you provide a more complete definition of "obscurity" (e.g. end of 3.1)? This term sounds somewhat subjective.

Q4: I was confused about the claim in 5.3 that the alpha scores are not interpretable but the beta scores are. Why can't we meaningfully interpret alpha as a measurement of benchmarking strength? The paper says it is influenced by the "prompts and specifics of the implementation", but if that is true why wouldn't that argument also apply to beta?

Q5: Could you comment on how you imagine this could be applied to any domains besides ones that require mathematical proofs? For example, as mentioned earlier, I'm not convinced that this would be so easy even in the domain of programming, as finding unit test cases that refute an implementation may be much harder than implementing something that almost works. I'd be curious to hear what other domains might be suitable as well, and what that might look like.

Q6: A typical criticism of adversarial benchmarking is that it results in an unnatural distribution of questions. Similarly, here question generators are incentivized to ask challenging questions, which may lead them to choose questions about obscure topics or are otherwise very different from real use-cases. For instance, in the math domain a generated problem could ask a model to prove some property of some very strange function that no real mathematician cares about. Do you think this is a real concern? What can be done to mitigate it?

Q7: In 4.1 it sounds like the feasibility gating is performed by a (benchmarker, answerer) pair. However in 6.1 under "Dataset construction", there is one question for each (model, category) pair. How is the feasibility gating done then? I can imagine two possibilities: (1) A question is feasible only if the benchmarker's answer passes critique from every answerer, or (2) A question is feasible or infeasible only in the context of a particular answerer. Thus, a question could be considered feasible when posed to one answerer, but infeasible when posed to another. The latter sounds somewhat strange (a question ought to be either feasible or infeasible), but perhaps this edge case does not happen too often.

**Limitations:**

yes

**Strengths And Weaknesses:**

First, the overall idea of the paper is innovative. Overturning the standard paradigm of benchmarking where humans provide correct answers is an ambitious goal, and the paper provides a well-designed alternative. In particular, the paper insightfully leverages the relative tractability of verifying alleged errors in domains such as mathematics and computer science.

The paper is also written well, with a lot of thoughtful argumentation for why the proposed benchmarking paradigm is needed.

The biggest weakness of the paper is that the experimental portion is quite sparse. The paper presents one main experiment implementing this benchmarking paradigm with a set of 8 LLMs. They extract rankings of these models and find that they correlate well with existing math benchmarking results. However, this finding by itself is quite a low bar. I imagine that one could design many benchmarks that correlate with the generally accepted strengths of these models. Besides these results, the only other results verify that even weak models agree with human adjudicators on evaluating local witnesses.

I would suggest many additional analyses of the experiments, including:
* Some examples of generated questions and qualitative analysis of the questions
* One big question that remains is, are there cases where the answer is actually wrong but is judged correct because the error was not found by the debate process? This seems to be an important question for determining the validity of the proposed paradigm. The gold standard here would be to have human experts judge the entire answers and then compare their verdicts with the outcomes from the benchmark. I understand that the paradigm envisions a future where humans cannot do this sort of evaluation, but I don't think we are there quite yet (or maybe we are, but it's hard for me to say without seeing the generated questions for this benchmark).
* One heuristic way to get at this same question would be to compare the answers that different models provided to the same question, and check if there were any cases where two models produced answers that were judged correct, but the answers are mutually incompatible.
* Are the alpha and beta scores highly correlated? Are there models that are better at answering than benchmarking, or vice versa?
* Since the paper mentioned that question generation is sensitive to prompting choices (5.3), the experiments should analyze the sensitivity of the results to these prompting choices. The only robustness checks performed in the experiments use bootstrapping, which is much weaker and only tests robustness to random sampling, not design decisions.

Another weakness is that the domains in which this type of benchmarking can be applied are quite limited. Besides mathematics, I am not convinced that it would be successful in another domain. For instance, the paper mentions computer science, where unit tests serve as  witnesses of incorrectness, but I can imagine that a model could generate code that is incorrect but for which it may be very difficult to find a unit test exposing the problem.

Edit: After the author response period, I am raising my score as they have addressed my concerns adequately.

---

> ### Author Rebuttal · Authors · 2026-03-30
>
> Dear Reviewer Uovv,
>
> Thank you for your detailed review and the experimental suggestions\!
>
> We ran all of the analyses you proposed (at an additional cost of ~$1k USD), and we believe the results address your core concern about experimental depth. Due to the character limit, we’ll have to be a bit brief, so please check out our other rebuttals as well.
>
> **New Experimental Results**
>
> *Question examples* See Reviewer 8ag1’s rebuttal for an example + links. We’ll also include samples in the Appendix with a qualitative analysis.
>
> *False negatives.* Since full human evaluation of every answer is infeasible, we used GPT-5.4-high (which is substantially stronger than all evaluated models) as a ground-truth proxy. Our key finding is that critics have correctness rates that are uniformly high across all models (0.88-0.96 F1), while answer correctness rates vary enormously (0.16-0.99). In other words, even weak models are reliable error-spotters; it is much harder to produce correct answers than to catch mistakes in them. This asymmetry is what makes the protocol work.
>
> *Mutual incompatibility.* Following your suggestion, we checked how often two non-proven-wrong answers to the same question were mutually incompatible (classified by GPT-5.4-high). The overall rate is 4.23% (160/3779 pairs). Crucially, this is concentrated in pairs involving weak models: GPT-3.5 pairs show 14-18% incompatibility, while frontier model pairs (Claude Opus 4.5, DeepSeek v3.2, GPT-5.2, Gemini 3 Pro) show 0-0.4%.
>
> *Prompt sensitivity.* We ran a prompt ablation with 2 benchmarkers, 7 answerers, and 3 prompt variants (standard, no guidance document, no knowledge of self-check), using GPT-5.4-high as evaluator. We find that the $\beta$ rankings are highly stable: Kendall $W$ = 0.952, mean pairwise Spearman $\rho$ = 0.929, and mean pairwise Kendall $\tau$ = 0.873. In other words, the protocol is robust to prompt differences.
>
> *Alpha-beta correlation.* Benchmarker and answerer scores are moderately but not fully correlated (Kendall $\tau$ = 0.47, Spearman $\rho$ = 0.69), which is interesting! For example, Llama 4 Maverick is a decent answerer but a weak benchmarker.
>
> You can find our raw data (anonymously) [here](https://pastebin.com/xYX88uU7).
>
> **Responses to Questions**
>
> *Q1 (Tripartite decomposition).* We explored exactly this tripartite decomposition (Questioner, Answerer, Checker) and abandoned it due to an identifiability problem: it’s impossible to reliably distinguish between “the question is too easy” and “the critic didn’t spot the mistake”, so we merged Questioner and Checker into Benchmarker. See our response to Reviewer aQbn for the full discussion.
>
> *Q2 (Comprehension wording).* We agree: "impractical" is more precise than "impossible," and only the weaker form is needed for R2. We will soften the wording.
>
> *Q3 (Obscurity).* An answer is obscure if critical steps are missing or unexplained such that correctness cannot be verified within reasonable effort. In practice, no obscurity claims arose in our experiments, though we kept the category for the sake of future-proofing.
>
> *Q4 (Alpha interpretability).* We agree that both $\alpha$ and $\beta$ are prompt-dependent in principle, although answering is a more constrained task (a definite question with a definite answer), and consistent answer performance is something explicitly optimized for. The reason we minimized the importance of $\alpha$ is that the interpretation of high or low $\alpha$ is less direct than low or high $\beta$, even though the new experiments confirm that $\alpha$ is fairly robust. We will soften the claim in the camera-ready.
>
> *Q5 (Domains beyond math).* We are extending CRB to coding with promising early results. See our response to Reviewer 8ag1 for details.
>
> *Q6 (Unnatural question distribution).* This is a real concern. We mitigate it by constraining topics via MSC2020 categories and allowing answerers to flag questions as ill-posed. We originally tested an earlier protocol version that included a human meaningfulness check (where humans would label if the question was meaningfully measuring math capabilities), but we abandoned it for epistemic reasons (how can humans objectively evaluate meaningfulness?). This could be reintroduced with clearer meaningfulness criteria.
>
> *Q7 (Feasibility gating).* We follow approach (1): a question is feasible only if the benchmarker's self-answer passes critique from all other models. We will clarify this.
>
> We hope that these results address your concerns. We have conducted every analysis you suggested, and in each case, the results support the protocol's validity. Thank you, as this new data makes the paper much stronger. Combined with the theoretical contributions, we believe that the paper makes a strong contribution and hope you will consider revisiting your score in light of these new results. Let us know if you have further questions or if there are other analyses you’d like us to run.
>
> Thank you again!
>
> The Authors

---

> > ### Author Rebuttal · Reviewer_Uovv · 2026-04-04
> >
> > Thank you for your detailed and clear answers to my questions! I will raise my score to 5.

---

> > > ### Author Response · Authors · 2026-04-04
> > >
> > > Thank you so much!
> > >
> > > The Authors

---

### Official Review · Reviewer_8ag1 · 2026-03-20

**Soundness:** 4
**Presentation:** 3
**Significance:** 4
**Originality:** 4
**Overall Recommendation:** 5
**Confidence:** 3

**Summary:**

This paper introduces Critique-Resilient Benchmarking (CRB) for benchmarking in settings where model capabilities surpass human performance to the point that humans can no longer reliably generate challenging questions, verify ground-truth answers, holistically evaluate solutions, and assign meaningful difficulty labels to individual questions. The core idea is to replace grading against a ground-truth answer with a notion of performance called critique-resilient correctness: an answer is considered correct if adversarial critics cannot produce a valid localized proof of error or ill-posedness under compute-bounded verification by a third-party. The full protocol includes a feasibility gate in which the benchmarker proposes a question and answer that itself survives critique to avoid ill-posed questions, surviving questions are answered by models and these answers are critiqued by critics, and outcomes of this process are used to fit an itemized bipartite Bradley–Terry model to estimate both answerer and benchmarker strength. The main experiment in this paper studies this protocol  in the domain of mathematics problems and proofs. They report that protocol produces stable rankings under repeated trials, correlation rankings with external math benchmarks, and similar rankings when weak models act as adjudicators.

**Compliance With Llm Reviewing Policy:**

Affirmed.

**Final Justification:**

Keeping my score as is!

**Key Questions For Authors:**

How robust is CRB to strategic gaming by answerers? Is there any statement that can be made along the lines of "Optimizing for CRB performance is (or is not) always guaranteed to result in better more capable models?"
Feasibility gating requires the benchmarker to provide a self-answer that survives critique. To what extent does benchmarker strength therefore reflect answering strength, and is there a cleaner way to disentangle those two factors?
Beyond mathematics, which concrete domains do the authors believe satisfy the witness-admitting property strongly enough for CRB to remain reliable in practice?
How do the authors ensure that model-generated questions probe  target capabilities of interest rather than difficult-but-uninteresting capabilities?

**Limitations:**

Yes

**Strengths And Weaknesses:**

Strengths

Addresses an important question given the difficulty of developing lasting benchmarks in areas where models are highly performant.
Each component of the protocol directly follows from one of the four assumptions in Section 2.1, and these assumptions are very reasonable given the state of current benchmarking.
The paper is explicit about the soundness/completeness tradeoff and does not overclaim that critique-resiliency implies true correctness.
The paper goes beyond a purely conceptual proposal and performs an empirical analysis of this protocol in a realistic setting.


Weaknesses

The paper could benefit from discussion of how CRB systems hold up under  adversarial model trainers. Currently benchmarks have a large role in the AI research community to the point that many model providers specifically target certain benchmarks as these are our main measures of progress (i.e. "benchmaxxing"). One possibility is that adopting benchmarks based on CRB yields models that are highly proficient at convincing compute-bounded critics rather than truly improving their underlying capabilities.
As per my reading of Section 4.1, the feasibility gate requires that the benchmarker itself produces a critique-resilient answer. If this is the case, does this not imply that only the strongest answerer can produce benchmarks that other models cannot saturate? For instance, if a relatively bad answerer is used as a benchmarker, then either (1) their proposed questions will get gated due to incorrect self-answers or (2) their  questions will easily be answered by stronger answerers (given that the relatively weak answerer gave a critique-resilient answer). I may be misunderstanding the protocol here, but if so I think this needs clarification in the paper. If this understanding is correct, then this is a pretty significant weakness of the protocol.
Benchmarks are usually designed to test specific competencies. How does one ensure that a benchmarker remains faithful to testing the specific competency of interest. One failure mode I could imagine is that a benchmarker could start to produce questions that are difficult, but not very interesting. For instance, in math this might be problems that require extremely complicated algebra -- can be difficult to solve but is not necessarily an interesting type of difficulty.
The empirical evaluation is limited to mathematics, which seems like a particularly favorable witness-admitting domain, so generalization to other domains remains unclear. It would be nice to discuss other domains that both are and are not witness-admitting.

---

> ### Author Rebuttal · Authors · 2026-03-30
>
> Dear Reviewer 8ag1,
>
> Thank you for your thoughtful and detailed review\! We are glad that you highlight the importance of the problem we address, the soundness of our formalization, and the strength of our empirical validation. We now address each concern you raised.
>
> **1. Robustness to Strategic Gaming**
>
> Indeed, any adversarial evaluation (and, to a certain extent, any evaluation at all) is at risk of gaming. That said, in witness-admitting domains, a critic only needs to find one valid witness to succeed, and as the critic pool grows in strength and diversity, the set of incorrect-but-unfalsifiable answers shrinks. The obscurity mechanism provides an additional safeguard: since opaque solutions count as answerer failures, the answers must be easy to evaluate, or they are automatically rejected.
>
> To provide more empirical evidence of the answering vs critiquing gap, we ran new analyses using GPT-5.4 (which is much stronger than the studied models) as an approximation of the ground truth, and found that the answerer accuracy is 65.1%, while the average critic accuracy is 93.4% (F1: 0.965). See the full results [here](https://pastebin.com/xYX88uU7). In other words, it’s easier to catch errors than to avoid making them, which suggests that honest answering can be the dominant strategy.
>
> Still, your concern remains valid in principle: an answerer could optimize for the boundary of what current critics detect. As part of follow-up work, we are exploring an extension where answers later falsified by stronger models receive harsher penalties in the BT model.
>
> So to answer your question: the evidence points to optimizing for CRB leading to more capable models, but more research is needed. Also, an advantage of CRB is that traditional benchmarks face the same gaming dynamics with far less transparency, while CRB makes the failure mode auditable.
>
> **2. Benchmarker-Answerer Correlation**
>
> Our data shows that benchmarker and answerer Elo are moderately but not fully correlated (Spearman $\rho$ = 0.69, Kendall $\tau$ = 0.47): for example, GPT-4o and GPT-3.5 rank 5th and 6th as benchmarkers, while Llama 4 Maverick, a stronger answerer, drops to 7th (see Figure 5 in the Appendix for the benchmarker Elo scores). In other words, generating question-answer pairs and answering other models’ questions are related but distinct capabilities (prime factorization is an extreme example of this phenomenon: it’s very easy to generate as a question-answer pair but infeasible to answer on its own).
>
> **3. Question Quality**
>
> There are three aspects of our design that mitigate this concern:
>
> * We use the MSC2020 categories as seeds to enforce variety across all mathematical domains.
> * Answerers can flag computationally heavy problems as ill-posed.
> * We manually inspected the generated questions and found them to be both high-quality and far from trivial (they can be inspected anonymously [here](https://anonymous.4open.science/r/critique-resilient-benchmarking/benchmarks/anthropic-claude-opus-4.5.json)). We report an example question by GPT-5.2 (topic: Dynamical Systems and Ergodic Theory):
>
> > Let $\mathbb T^2:=\mathbb R^2/\mathbb Z^2$ be the 2-torus with Haar (Lebesgue) probability measure (m).
> > Let $$ A=\begin{pmatrix}2&1 \ | \ 1&1\end{pmatrix}\in SL(2,\mathbb Z), \qquad T:\mathbb T^2\to\mathbb T^2,\quad T([x])=[Ax], $$ where $[x]$ denotes the class of $x\in\mathbb R^2$ modulo $\mathbb Z^2$.
> > 1. Prove that $m$ is $T$-invariant.
> > 2. Prove that $T$ is mixing with respect to $m$, i.e. for all $f,g\in L^2(\mathbb T^2,m)$, $$ \int_{\mathbb T^2} f\circ T^n , g , dm ;\longrightarrow; \int f,dm \int g,dm \quad\text{as }n\to\infty. $$
> > 3. Compute the Kolmogorov-Sinai entropy $h_m(T)$ exactly.
>
> We’ll add samples in the Appendix.
>
> **4. Generalization Beyond Mathematics**
>
> As part of follow-up work, we are extending CRB to other tasks:
>
> * Algorithmic problem-solving: the benchmarker designs a task with formalized output constraints (think LeetCode), the answerer writes a solution, and the benchmarker finds a counterexample input.
> * Mechanism design: the benchmarker proposes a strategic setting (e.g. an auction), the answerer designs a protocol, and the benchmarker finds an adversarial strategy.
> * Program rewriting: the benchmarker writes code in one language, the answerer rewrites it, the benchmarker identifies semantic differences.
>
> We expected algorithmic problem-solving to be saturated and easy to check for humans, but in practice CRB has been very helpful as many solutions are beyond even some CS PhDs’ skills. For the other two, we’re still conducting exploratory work, but the early results are promising.
>
> We appreciate the depth of your review, and we will expand the discussion of gaming robustness and benchmarker-answerer disentanglement in the camera-ready version. We hope that these responses increase your confidence in the quality of our paper, and let us know if you have any other questions.
>
> Thank you again!
>
> The Authors

---

> > ### Author Rebuttal · Reviewer_8ag1 · 2026-04-06
> >
> > Thank you!!

---

### Decision · Program_Chairs · 2026-04-30

**Decision:**

Accept (spotlight)

**Comment:**

This paper aims to address the benchmarking challenge of frontier models when their capabilities are infeasible or impractical for humans to directly evaluate. They focused on math reasoning ad proposed a gamified strategy between two models, a benchmarker and an answerer. The benchmarker proposes a question, and the answerer answers it. After this, the benchmarker gives a critique consisting of a short witness that is supposed to invalidate the answerer's solution. At the end of this game, the benchmarker wins if they produce a feasible question which the answerer fails to provide a solution to that survives critique. If the solution does survive, then the answerer wins. They collect the pairwise win rate data and derive the Elo score out of them using a Bradley-Terry Model. The main idea depends on critique-resilient correctness (CRC): an answer is deemed correct if no adversary has convincingly proved otherwise. The main advantage is that human oversight is not required to check the entire answer (which could be a long mathematical proof), but only to check the local flaws identified during this later verification process.

Reviewers agree that the paper addresses an important, realistic challenge for frontier model evaluation. The idea is innovative, and the assumptions about CRC are reasonable. The presentation and writing are also excellent to help people understand the method well. On the weaknesses side, reviewers found that the experiments are weak to some extent. They also requested additional analyses. Reviewers also required discussions about robustness to strategic gaming, the consistency between the evaluation scores of the two roles, etc. The rebuttal from authors addressed most of these concerns and reviewers are satisfied with them. Therefore, I recommend acceptance of this paper.